# Beyond Squared Error: Exploring Loss Design for Enhanced Training of Generative Flow Networks

**Rui Hu**[*], **Yifan Zhang**[*], **Zhuoran Li**, **Longbo Huang**[†]
Institute for Interdisciplinary Information Sciences, Tsinghua University
{hu-r24,zhangyif21,lizr20}@mails.tsinghua.edu.cn
longbohuang@tsinghua.edu.cn

## Abstract

Generative Flow Networks (GFlowNets) are a novel class of generative models designed to sample from unnormalized distributions and have found applications in various important tasks, attracting great research interest in their training algorithms. In general, GFlowNets are trained by fitting the forward flow to the backward flow on sampled training objects. Prior work focused on the choice of training objects, parameterizations, sampling and resampling strategies, and backward policies, aiming to enhance credit assignment, exploration, or exploitation of the training process. However, the choice of regression loss, which can highly influence the exploration and exploitation behavior of the under-training policy, has been overlooked. Due to the lack of theoretical understanding for choosing an appropriate regression loss, most existing algorithms train the flow network by minimizing the squared error of the forward and backward flows in log-space, i.e., using the quadratic regression loss. In this work, we rigorously prove that distinct regression losses correspond to specific divergence measures, enabling us to design and analyze regression losses according to the desired properties of the corresponding divergence measures. Specifically, we examine two key properties: zero-forcing and zero-avoiding, where the former promotes exploitation and higher rewards, and the latter encourages exploration and enhances diversity. Based on our theoretical framework, we propose three novel regression losses, namely, Shifted-Cosh, Linex(1/2), and Linex(1). We evaluate them across three benchmarks: hyper-grid, bit-sequence generation, and molecule generation. Our proposed losses are compatible with most existing training algorithms, and significantly improve the performances of the algorithms concerning convergence speed, sample diversity, and robustness.

## 1 Introduction

Generative Flow Networks (GFlowNets), introduced by Bengio et al. (2021; 2023), represent a novel class of generative models. They have been successfully employed in a wide range of important applications including molecule discovery (Bengio et al., 2021), biological sequence design (Jain et al., 2022), combinatorial optimization (Zhang et al., 2023), and text generation (Hu et al., 2024), attracting increasing interests for their ability to generate a diverse set of high-quality samples.

GFlowNets are learning-based methods for sampling from an unnormalized distribution. Compared to the learning-free Monte-Carlo Markov Chain (MCMC) methods, GFlowNets provide an alternative to exchange the complexity of iterative sampling through long chains for the complexity of training a sampler (Bengio et al., 2023). GFlowNets achieves this by decomposing the generating process into multiple steps and modeling all possible trajectories as a directed acyclic graph (DAG). The training goal is to determine a forward policy on this DAG, ensuring that the resulting probability distribution over terminal states aligns with the unnormalized target distribution. However, achieving this alignment is challenging due to the necessity of marginalizing the forward policy across a vast

---

[*]Equal contribution
[†]Corresponding author

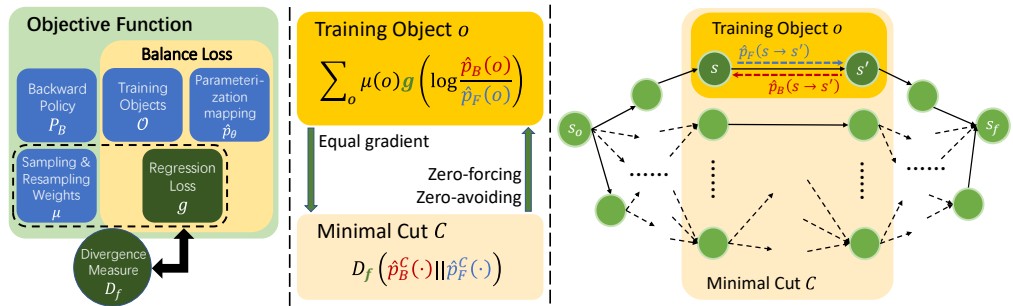

Figure 1: An illustration of our main theoretical results: the unified framework for GFlowNet training algorithms and the correspondence between regression losses over forward and backward flows on training objects and $f$-divergences between the two flows over minimal cuts.

trajectory space. To address this, GFlowNets utilize a backward flow to distribute the unnormalized target distribution over trajectories, thereby aligning the forward and backward flows.

Building on this foundation, various training algorithms for GFlowNets have been proposed, aiming to enhance the training of GFlowNets from different aspects such as credit assignment, exploration, and exploitation. Depending on the main focus of the methods, these algorithms can be divided into four categories, including training objects (Malkin et al., 2022; Madan et al., 2023), parameterization methods (Pan et al., 2023a; Deleu et al., 2022), sampling and resampling strategies (Rector-Brooks et al., 2023; Kim et al., 2024d; Lau et al., 2024) and the selection of backward policies (Shen et al., 2023; Jang et al., 2024a).

Most existing algorithms train the flow network by minimizing the squared error of the forward and backward flows in log-space, i.e., using the quadratic regression loss. However, there may exist more potential choices for loss functions beyond square error. Intuitively, any convex function that is minimized at zero point also provides a valid objective, in the sense that the forward and backward policies are aligned if and only if the loss is minimized. Further, the gradients of different regression losses lead to different optimization trajectories of the forward policy, thus highly influencing the exploration and exploitation behaviors. Yet, due to the lack of theoretical understanding for choosing an appropriate regression loss, it is unclear whether the above intuition is practical. In particular, the following central question remains open:

*Can a theoretical foundation be established for designing and analyzing regression loss functions?*

To answer this question, we conduct a systematic investigation of the largely overlooked regression loss aspect in GFlowNet training. Specifically, building on the work of Malkin et al. (2023), which established that training GFlowNets is analogous to optimizing a KL divergence, we rigorously prove that the gradient of the objective function using different regression losses, when combined with appropriate proposal distributions and resampling weights, equal to that of distinct divergence measures between the target distribution and the flow network-induced distribution. As different divergence measures endow the training objectives with different properties, and hence show different characteristics in the training process, our results provide a unified framework to generalize existing training methods and provide a principled way of designing efficient regression losses for GFlowNets training. Fig. 1 provides an overview of our technical results.

In particular, we study two important properties of the training objectives, i.e., **zero-forcing** and **zero-avoiding**, and systematically investigate their effects. In general, zero-forcing losses encourage exploitation, while zero-avoiding losses encourage exploration. Equipped with our new framework, we design three novel regression losses, namely **Linex(1)**, **Linex(1/2)**, and **Shifted-Cosh**, filling the four quadrants made up of the zero-forcing and zero-avoiding properties. We evaluate the new losses on three popular benchmarks: hyper-grid, bit-sequence generation, and molecule generation. Our results show that the newly

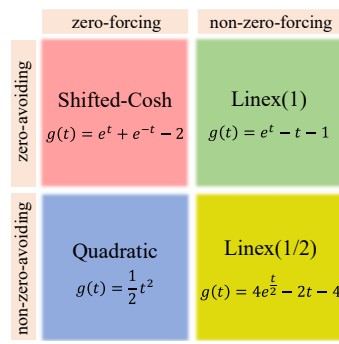

Figure 2: Our proposed regression losses and their properties.

proposed losses exhibit significant advantages over existing losses in terms of diversity, quality, and robustness, demonstrating the effectiveness of our design framework.

Our contributions can be summarized as follows:

- We develop a novel framework of the objective functions for training GFlowNets. This new framework identifies five key components of the objective function and unifies existing GFlowNet training algorithms including Flow-Matching GFlowNets (Bengio et al., 2021), Detailed-Balance GFlowNets (Bengio et al., 2023), Trajectory-Balance GFlowNets (Malkin et al., 2022), Sub-Trajectory-Balance GFlowNets (Madan et al., 2023) and their variants like Forward-Looking GFlowNets (Pan et al., 2023b) and DAG GFlowNets (Deleu et al., 2022) , etc (see Section 4.1).
- We establish the correspondence between the various objective functions for GFlowNets and different divergence measures. This insight facilitates a deeper understanding of how to design and analyze effective training objectives for GFlowNets (See Section 4.2).
- Based on our framework, we conduct an in-depth investigation on two key properties of regression losses: zero-forcing and zero-avoiding. We then design three new loss functions possessing different exploration/exploitation features, namely, Linex(1), Linex(1/2), and Shifted-Cosh (see Section 4.3).
- We conduct extensive experiments on three popular benchmarks: hyper-grid (Bengio et al., 2021), bit-sequence generation (Malkin et al., 2022), and molecule generation (Bengio et al., 2021). Our results demonstrate that the new losses significantly outperform the common squared loss in metrics including convergence speed, diversity, quality, and robustness (see Section 5).

## 2  RELATED WORK

**Generative Flow Networks (GFlowNets).** GFlowNets were initially proposed by Bengio et al. (2021) for scientific discovery (Jain et al., 2023a) as a framework for generative models capable of learning to sample from unnormalized distributions. The foundational theoretical framework was further developed by Bengio et al. (2023). Since then, numerous studies have focused on enhancing GFlowNet training from various perspectives, such as introducing novel balance conditions and loss functions (Malkin et al., 2022; Madan et al., 2023), refining sampling and resampling strategies (Shen et al., 2023; Rector-Brooks et al., 2023; Kim et al., 2024d; Lau et al., 2024), improving credit assignment (Pan et al., 2023a; Jang et al., 2024b) and exploring different options for backward policies (Shen et al., 2023; Mohammadpour et al., 2024; Jang et al., 2024a). Notably, our proposed method is compatible with all the aforementioned works as we have identified a novel key component of the objective functions for training GFlowNets.

From a theoretical perspective, GFlowNets are closely related to variational inference (VI, Malkin et al. 2023; Zimmermann et al. 2023) and entropy-regularized reinforcement learning (RL) on deterministic MDPs (Tiapkin et al., 2024; Mohammadpour et al., 2024). All of them can be viewed as solving distribution matching problems, and the gradients of their training objectives are equivalent to that of the reverse KL divergence. Our main theorem (Theorem 4.1 and Theorem B.1) generalizes the theoretical results of Malkin et al. (2023) by (i) extending reverse KL to the whole family of $f$-divergence and (ii) including more parameterization methods.

People also try to extend the formulation of GFlowNets to more complex scenarios, allowing continuous space (Lahlou et al., 2023), intermediate rewards (Pan et al., 2023b), stochastic rewards (Zhang et al., 2024b), implicit reward given by priority (Chen & Mauch, 2024), conditioned rewards (Kim et al., 2024c), stochastic transitions (Pan et al., 2023c), non-acyclic transitions (Brunswic et al., 2024), etc. Equipped with these techniques, GFlowNets are applied to an increasingly wide range of fields including molecular discovery (Jain et al., 2023b; Zhu et al., 2024; Pandey et al., 2024), biological sequence design (Jain et al., 2022; Ghari et al., 2023), causal inference (Zhang et al., 2022; Atanackovic et al., 2024; Deleu et al., 2024), combinatorial optimization (Zhang et al., 2023; Kim et al., 2024b), diffusion models (Zhang et al., 2024a; Venkatraman et al., 2024) and large language models (Hu et al., 2024; Song et al., 202Fani4).

**Divergence measures in training generative models.**  The properties of divergence measures and their effects as training objectives have been studied by Minka et al. (2005). As the original training objectives of many generative models are equivalent to one of the divergence measures (typically the reverse KL divergence), it is natural to introduce a more general class of divergence measures to

replace it. This idea has successfully improved the performances of a variety of algorithms for training generative models, including GAN (Nowozin et al., 2016; Arjovsky et al., 2017), VAE (Zhang et al., 2019), VI (Li & Turner, 2016; Dieng et al., 2017), Distributional Policy Gradient (DPG for RL, Go et al. 2023), and Direct Preference Optimization (DPO for RLHF, Wang et al. 2024). A very recent study (Silva et al., 2024) also explores this idea in the context of GFlowNets by investigating the use of four different divergence measures: forward KL, reverse KL, Renyi-$\alpha$ and Tsallis-$\alpha$.

In contrast to the studies mentioned above, this work establishes a two-way connection between $f$-divergences and the regression loss function $g$ in the training objectives of GFlowNets. By examining the zero-forcing and zero-avoiding properties of these divergence measures, we can opt for the desired regression loss to enhance exploration and/or exploitation for training GFlowNets.

## 3  PRELIMINARIES OF GFLOWNETS AND $f$-DIVERGENCE

In this section, we first present preliminaries of GFlowNets and the $f$-divergence, which will be the foundation of our subsequent exposition.

### 3.1  GFLOWNETS

A GFlowNet is defined on a directed acyclic graph $G = (V, E)$ with a source node $s_o$ and a sink node $s_f$, such that every other vertex is reachable starting from $s_o$, and $s_f$ is reachable starting from any other vertex. Let $\mathcal{T}$ be the collection of all complete trajectories, and $\Sigma$ be the corresponding $\sigma$-algebra, then a **flow** is a measure $F$ on $(\mathcal{T}, \Sigma)$.

Further, we define **state-flow**, **edge-flow** and **total flow** by

$$F(s) := F(\{\tau : s \in \tau\}), \quad F(s \to s') := F(\{\tau : (s \to s') \in \tau\}), \quad Z := F(\{\mathcal{T}\}).$$

A flow then induces a forward probability $P_F(s'|s)$ and a backward probability $P_B(s|s')$, defined as:

$$P_F(s'|s) := P(s \to s'|s) = \frac{F(s \to s')}{F(s)}, \quad P_B(s|s') := P(s \to s'|s') = \frac{F(s \to s')}{F(s')}.$$

**Markovian flow** is a special family of flows such that at each step, the future behavior of a particle in the flow stream only depends on its current state. Formally speaking, let $\iota$ be any trajectories from $s_o$ to $s$, then $P(s \to s'|\iota) = P(s \to s'|s) = P_F(s'|s)$. We focus on Markovian flows in the following.

A set of (not necessarily complete) trajectories $C$ is a **cut** if and only if for any complete trajectory $\tau$, there exists $\iota \in C$ such that $\iota$ is a part of $\tau$. Here we view vertices and edges as trajectories of length 1 or 2 and further extend the definition of $F$ to all trajectories as

$$F(\iota) = F(\{\tau : \iota \text{ is a part of } \tau\}).$$

A **minimal cut** is a cut such that the sum of flows in the cut is minimized. According to the max-flow min-cut theorem, this amount is equal to $Z$ the total flow. Let $\mathcal{C}$ be the collection of all minimal cuts, then for each minimal cut $C \in \mathcal{C}$, let $p^C(\iota) := F(\iota)$ for all $\iota \in C$, then $p^C(\cdot)$ can be viewed as an unnormalized distribution over $C$.

Let the terminating set $S^f$ be the collection of nodes that directly link to $s_f$. Note that $C = \{(s \to s') : s' = s_f\}$ is a minimal cut, so $p^C(\cdot)$ induces a distribution over $S_f$. We denote it as $p_F^T$ and its induced probability distribution as $P_T$ (called the **terminating probability**):

$$\forall s \in S^f, p_F^T(s) = F(s \to s_f), \quad P_T(s) = \frac{p_F^T(s)}{\sum_{s' \in S^f} p_F^T(s')} = \frac{F(s \to s_f)}{Z}.$$

The ultimate goal of training a GFlowNet is to match $p_F^T$ with $R$, so that the forward policy draws samples from $P_T = P_R$, where $P_R$ denotes the normalized probability distribution defined by $R$.

### 3.2  $f$-DIVERGENCE

The $f$-divergence is a general class of divergence measures (Liese & Vajda, 2006; Polyanskiy, 2019):

$$D_f(p||q) = \sum_{x \in \mathcal{X}} q(x) f\left(\frac{p(x)}{q(x)}\right) + f'(\infty) p(\{x \in \mathcal{X} : q(x) = 0\}),$$

where $p$ and $q$ are two probability distributions on a measurable space $(\mathcal{X}, \mathcal{F})$, $f : \mathbb{R}_{++} \to \mathbb{R}$ is a twice differentiable convex function with $f(1) = f'(1) = 0$ , and $f'(\infty) = \lim_{t \to +\infty} \frac{f(t)}{t}$. Hence, the Kullback-Leibler (KL) divergence (Zhu & Rohwer, 1995) $D_{\mathrm{KL}}(p||q)$ and $D_{\mathrm{KL}}(q||p)$ correspond to $D_f(p||q)$ with $f(t) = t \log t - t + 1$ and $f(t) = t - \log t - 1$, respectively. When $f(t) = -\frac{t^\alpha}{\alpha(1-\alpha)} + \frac{t}{1-\alpha} + \frac{1}{\alpha}$, the $f$-divergence corresponds to the $\alpha$-divergence $D_\alpha(p||q)$ introduced in (Zhu & Rohwer, 1995; Amari, 2012).

The $f$-divergence preserves the following nice properties of KL divergence, ensuring that they can also serve as good optimization objectives.

**Fact 3.1** (Liese & Vajda (2006)). $D_f(p||q) = 0$ if and only if $p = q$.

**Fact 3.2** (Liese & Vajda (2006)). $D_f(p||q)$ is convex with respect to either $p$ or $q$.

The definition of $D_f(p||q)$ can be further extended to all twice differentiable functions $f$ with $f(1) = f'(1) = 0$, termed pseudo $f$-divergence.

## 4 TRAINING GENERATIVE FLOW NETWORKS

In this section, we present our perspective on analyzing GFlowNet training algorithms in detail. In Section 4.1, we provide a general framework with five customizable components to unify existing training algorithms. In Section 4.2, we dive deep into the regression loss component, which has been overlooked in existing research, and establish a rigorous connection between it and divergence measures. In Section 4.3, we further show how to utilize this connection for designing and analyzing objective functions.

### 4.1 A UNIFIED FRAMEWORK FOR GFLOWNET TRAINING ALGORITHMS

Consider the following general objective function for forward policy:

$$\mathcal{L}_{\mathcal{O}, \hat{p}_\theta, \mu, P_B, g} = \sum_{o \in \mathcal{O}} \mu(o) g \left( \log \frac{\hat{p}_B(o; \theta)}{\hat{p}_F(o; \theta)} \right) \tag{1}$$

This formulation is defined by five key components. (i) The set of training objects $\mathcal{O}$, which can include states, transitions, partial trajectories, or complete trajectories. (ii) The parameterization mapping $\hat{p}_\theta$, which defines how the parameters of the flow network represent the forward flow $\hat{p}_F$ and the backward flow $\hat{p}_B$. (iii) The sampling and resampling weights $\mu$, which influence how training objects are sampled and weighted. (iv) The choice of backward policy $P_B$, which can be either fixed or learned. (v) The regression loss function $g$, ensuring that the forward and backward policies align when minimized.

Table 1: Summary of existing GFlowNet training algorithms and techniques.

| Design Component | Algorithms |
|---|---|
| Training Objects $\mathcal{O}$ and Parameterization Mapping $\hat{p}_\theta$ | FM-GFN (Bengio et al., 2021), DB-GFN (Bengio et al., 2023), TB-GFN (Malkin et al., 2022), STB-GFN (Madan et al., 2023), FL-GFN (Pan et al., 2023a), DAG-GFN (Deleu et al., 2022; Hu et al., 2024) |
| Sampling/Resampling Weights $\mu$ | PRT (Shen et al., 2023), TS-GFN (Rector-Brooks et al., 2023), LS-GFN (Kim et al., 2024d), QGFN (Lau et al., 2024), Genetic-GFN (Kim et al., 2024a) |
| Backward Policy $P_B$ | GTB (Shen et al., 2023), ME-GFN (Mohammadpour et al., 2024), PBP-GFN (Jang et al., 2024a) |
| Regression Loss $g$ | **Ours** |

While most GFlowNets training objectives are not explicitly written in this form, Equation (1) unifies all existing training objectives. Table 1 presents a categorization of existing algorithms according to the components they specify.

In previous literature, $g(t) = \frac{1}{2}t^2$ has been the only choice for regression loss, and the term $g \left( \log \frac{\hat{p}_B(o; \theta)}{\hat{p}_F(o; \theta)} \right) = \frac{1}{2} \left( \log \frac{\hat{p}_B(o; \theta)}{\hat{p}_F(o; \theta)} \right)^2$ is usually referred to as the balance loss. It is specified by

the training objects and parameterization mapping. Popular balance losses are flow-matching (FM) loss, detailed-balance (DB) loss, trajectory-balance (TB) loss sub-trajectory-balance (STB) loss, and their modified versions. For example, the objective function of on-policy TB loss with fixed uniform $P_B$ can be written as

$$\mathcal{L} = \sum_{\tau \in \mathcal{T}} \hat{P}_F(\tau; \theta) \frac{1}{2} \left( \log \frac{\hat{p}_B(\tau; \theta)}{\hat{p}_F(\tau; \theta)} \right)^2,$$

where for $\tau = (s_0 = s_o, s_1, s_2, \cdots, s_{T-1}, s_T = s_f)$, we have

$$\hat{p}_F(\tau; \theta) = \hat{Z}(\theta) \hat{P}_F(\tau; \theta) = \hat{Z}(\theta) \prod_{t=1}^{T} \hat{P}_F(s_t | s_{t-1}; \theta),$$

$$\hat{p}_B(\tau; \theta) = R(s_{T-1}) P_B(\tau) = R(s_{T-1}) \prod_{t=1}^{T-1} P_B(s_{t-1} | s_t) = R(s_{T-1}) \prod_{t=1}^{T-1} \frac{1}{\text{indegree}(s_t)}.$$

Please refer to Appendix A for the detailed correspondence of other algorithms under this unified framework.

## 4.2 THE INFORMATION-THEORETIC INTERPRETATION OF TRAINING OBJECTIVES

Based on our proposed framework, We establish a novel connection between the $g$ functions and the $f$-divergences. The result is summarized in Theorem 4.1 below.

**Theorem 4.1.** *Let $\theta_F$ be the parameters for forward policies. For each minimal cut $C \in \mathcal{C}$, the restrictions of both forward and backward flow functions on $C$ can be viewed as unnormalized distributions over it, denoted as $\hat{p}_F^C$ and $\hat{p}_B^C$, respectively.*

*If there exists $w : \mathcal{C} \to \mathbb{R}_+$ such that $\mu(o) = \hat{p}_F(o) \sum_{C \in \mathcal{C}, o \in C} w(C)$ for any $o \in \mathcal{O}$, then*

$$\nabla_{\theta_F} \mathcal{L}_{\mathcal{O}, \hat{p}_\theta, \mu, P_B, g} = \nabla_{\theta_F} \sum_{C \in \mathcal{C}} w(C) D_f(\hat{p}_B^C || \hat{p}_F^C), \text{ where } f(t) = t \int_1^t \frac{g'(\log s)}{s^2} ds. \quad (2)$$

The theorem states that the expected gradient of the objective function equals the gradient of a weighted sum of $f$-divergence over minimal cuts if the sampling and resampling weights $\mu$ on each training object $o$ equals the forward flow times the accumulated weights on minimal cuts consisting of $o$. For example, $w(C) = \mathbb{I}[C = \mathcal{T}]$ corresponds to TB GFlowNets using on-policy sampling. The detailed proof of Theorem 4.1 is provided in Appendix B. We also provide a thorough discussion about the interpretations of FM, DB, and subTB loss under this framework. Please see Appendix C for details.

Note that when $g(t) = \frac{1}{2} t^2$, i.e., the popular squared loss, we obtain $f(t) = t - \log t - 1$. Thus, $D_f$ is the reverse KL divergence, recovering the results in Malkin et al. (2023). Compared to existing work, Theorem 4.1 offers a general connection and applies to a wide range of algorithms shown in Table 1. Below, we conclude this section with the following two remarks on the connections between function $g$ and $f$.

*Remark* 4.2. Note that $f(1) = 0$, $f'(1) = g'(0)$, and $f''(t) = \frac{g''(\log t)}{t^2}$. If $g$ is twice differentiable, then $D_f$ is an $f$-divergence if and only if $g$ is convex and is minimized at zero point.

*Remark* 4.3. Solving for $g$ from $f(t) = t \int_1^t \frac{g'(\log s)}{s^2} ds$ and $g(0) = 0$ gives $g(t) = f(e^t) - \int_1^{e^t} \frac{f(s)}{s} ds$.

## 4.3 DESIGNING NEW REGRESSION LOSSES

Equipped with the connection established in Theorem 4.1, we now show how one can build upon it and design regression losses with two important properties: **zero-forcing** and **zero-avoiding**. A zero-forcing objective leads to a conservative result, while a zero-avoiding objective offers a diverse approximated distribution. As pointed out by previous studies (Minka et al., 2005; Go et al., 2023), zero-forcing property encourages exploitation, while zero-avoiding property encourages exploration. Therefore, a zero-avoiding loss may converge faster to a more diverse distribution, while a zero-forcing one may converge to a distribution with a higher average reward.

To this end, we first study the effect of using different divergence measures as optimization objectives.

**Proposition 4.4** (Liese & Vajda (2006))**.** *Denote* $f(0) = \lim_{t \to 0^+} f(t)$, $f'(\infty) = \lim_{t \to +\infty} \frac{f(t)}{t}$,

1. *Suppose* $f(0) = \infty$, *then* $D_f(p||q) = \infty$ *if* $p(x) = 0$ *and* $q(x) > 0$ *for some* $x$.
2. *Suppose* $f'(\infty) = \infty$, *then* $D_f(p||q) = \infty$ *if* $p(x) > 0$ *and* $q(x) = 0$ *for some* $x$.

In particular, $D_\alpha(p||q)$ for $\alpha \leq 0$, including reverse KL divergence, satisfies the first condition, while $D_\alpha(p||q)$ for $\alpha \geq 1$, including forward KL divergence, satisfy the second condition. Proposition 4.4 leads to the following results of approximating a distribution by using $f$-divergence.

**Proposition 4.5** (Liese & Vajda (2006))**.** *Let $S$ be a subset of the collection of distributions over $\mathcal{X}$. Let $\hat{p}_S \in \arg\min_{q \in S} D_f(p||q)$.*

1. *Zero-forcing: Suppose* $f(0) = \infty$, *then* $\hat{p}_S(x) = 0$ *if* $p(x) = 0$.
2. *Zero-avoiding: Suppose* $f'(\infty) = \infty$, *then* $\hat{p}_S(x) > 0$ *if* $p(x) > 0$.

Proposition 4.5 suggests that when $S$ does not cover the target distribution $p$, the best approximation may vary according to the divergence chosen as the objective.

Since the objective functions for GFlowNets with varying regression losses are closely related to different divergence measures, we similarly define their zero-forcing and zero-avoiding properties.

**Definition 4.6.** An objective function $\mathcal{L}$ for training GFlowNets is

1. **Zero-forcing**: if for any parameter space $\Theta$ and $\theta^* = \arg\min_{\theta \in \Theta} \mathcal{L}(\theta)$,

$$\forall s \in S^f : R(s) = 0 \implies \hat{P}_T(s; \theta^*) = 0,$$

2. **Zero-avoiding**: if for any parameter space $\Theta$ and $\theta^* = \arg\min_{\theta \in \Theta} \mathcal{L}(\theta)$,

$$\forall s \in S^f : R(s) > 0 \implies \hat{P}_T(s; \theta^*) > 0.$$

In such cases, we also say that the regression function $g$ itself is **zero-forcing** or **zero-avoiding**.

We then have the following theorem regarding the zero-forcing and zero-avoiding objective functions and regression losses of GFlowNets.

**Theorem 4.7.** *Let $\mathcal{L}$ be an objective function for training GFlowNets, whose regression loss $g$ corresponds to $D_f$ according to Theorem 4.1. If $D_f$ is zero-forcing, then $\mathcal{L}$ and $g$ are both zero-forcing. If $D_f$ is zero-avoiding, then $\mathcal{L}$ and $g$ are both zero-avoiding.*

Table 2: Four representative $g$ functions and their corresponding $f$-divergences. Quadratic loss corresponds to reverse KL-divergence or the $\alpha$-divergence with $\alpha \to 0$. Linex(1) corresponds to forward KL-divergence or the $\alpha$-divergence with $\alpha \to 1$. Linex(1/2) corresponds to Hellinger distance (Hellinger, 1909) or the $\alpha$-divergence with $\alpha = 0.5$. Shifted-Cosh corresponds to an $f$-divergence that is both zero-forcing and zero-avoiding.

| Loss | $g(t)$ | $f(t)$ | $f(0)$ | $f'(\infty)$ | Zero-forcing | Zero-avoiding |
|---|---|---|---|---|---|---|
| Quadratic | $\frac{1}{2}t^2$ | $t - \log t - 1$ | $\infty$ | $1$ | ✓ | |
| Linex(1) | $e^t - t - 1$ | $t \log t - t + 1$ | $1$ | $\infty$ | | ✓ |
| Linex(1/2) | $4e^{\frac{t}{2}} - 2t - 4$ | $2t - 4\sqrt{t} + 2$ | $2$ | $2$ | | |
| Shifted-Cosh | $e^t + e^{-t} - 2$ | $t \log t - \frac{t}{2} + \frac{1}{2t}$ | $\infty$ | $\infty$ | ✓ | ✓ |

According to Theorem 4.7, the quadratic regression loss $g(t) = \frac{1}{2}t^2$ is a zero-forcing regression loss and focuses on exploitation. Combined with Remark 4.3 that enables us to determine a $g$ from an arbitrary $D_f$, we can easily find regression losses with both, either or neither of the zero-forcing and zero-avoiding properties. Since these two properties are finally rooted in $f(0)$ and $f'(\infty)$, our framework allows us to directly design a desired $g$ loss from a desired $D_f$. This provides a systematic and principled way of designing regression losses. For example, to obtain a zero-avoiding loss that focuses on exploration, we can solve for $g$ from $f(t) = t \log t - t - 1$ the forward KL divergence, which gives $g(t) = e^t - t - 1$ the Linex(1) function(Garg et al., 2023). We also design Linex(1/2) and Shifted-Cosh. The former is neither zero-forcing nor zero-avoiding, while the latter is both zero-forcing and zero-avoiding (see Table 2). In addition, we have also derived another five novel losses, corresponding to the forward and backward $\chi^2$ distance, total variation, symmetric KL divergence, and Jensen-Shannon divergence, respectively Please see Appendix F for details.

Note that a convex function $g$ with $g(0) = g'(0) = 0$ ensures the whole objective function is valid in the sense that the target distribution is perfectly matched if and only if the objective function is minimized, regardless of the choices of backward policy, training objects, parameterization, and exploration strategies. We will also see that they preserve the zero-forcing and zero-avoiding properties well from the empirical results in the following section.

## 5 EXPERIMENTS

In this section, we consider four representative $g$-functions (Table 2) and evaluate their performances over Flow-matching GFlowNets, Trajectory-balance GFlowNets, Detailed-balance GFlowNets, and Sub-trajectory-balance GFlowNets, across different choices of backward policies and sampling strategies. We consider the following three popular benchmarks, hyper-grid, bit-sequence generation, and molecule generation. Although the sampling and resampling weights $\mu$ may not fully meet the conditions of Theorem 4.1, the effects of zero-forcing and zero-avoiding properties are significant, demonstrating great compatibility with existing algorithms.

### 5.1 HYPER-GRID

We first consider the didactic environment hyper-grid introduced by Bengio et al. (2021). In this setting, the non-terminal states are the cells of a $D$-dimensional hypercubic grid with side length $H$. Each non-terminal state has a terminal copy. The initial state is at the coordinate $x = (0, 0, \cdots, 0)$. For a non-terminal state, the allowed actions are to increase one of the coordinates by 1 without exiting the grid and to move to the corresponding terminal state.

The reward of coordinate $x = (x_1, \cdots, x_D)$ is given according to

$$R(x) = R_0 + R_1 \prod_{i=1}^{D} \mathbb{I}\left[\left|\frac{x_i}{H} - 0.5\right| > 0.25\right] + R_2 \prod_{i=1}^{D} \mathbb{I}\left[0.3 < \left|\frac{x_i}{H} - 0.5\right| < 0.4\right],$$

where $0 < -R_1 < R_0 \ll R_2$. Therefore, there are $2^D$ reward modes near the corners of the hypercube.

We conducted experiments in 4-dimensional and 5-dimensional grids with $H = 20, R_0 = 10^{-4}, R_1 = -9.9 \times 10^{-5}, R_2 = 1 - 10^{-6}$. The backward policy is learned using the same objectives as the forward policy. We use the forward policy to sample training objects. We plot the empirical $L_1$ errors between $P_T$ and $P_R$ in Figure 3. Additional details can be found in Appendix E.1.

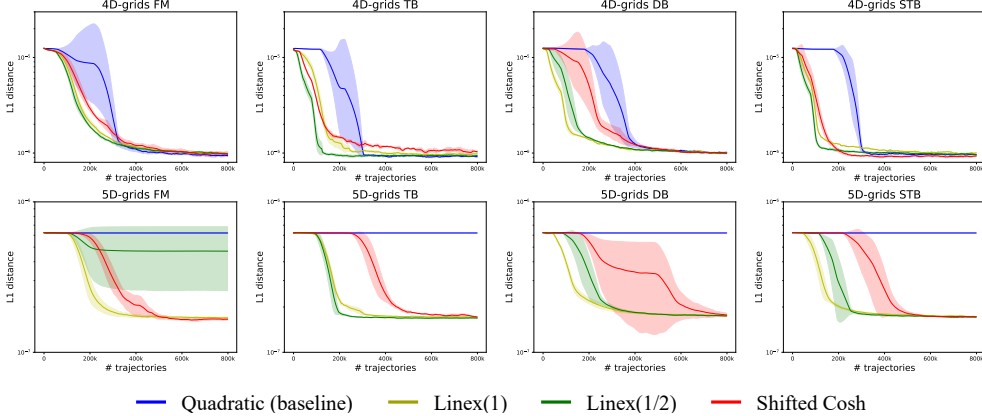

Figure 3: Hyper-grid results: the empirical L1 distance between $P_T$ and $P_R$.

As shown in Figure 3, the quadratic loss (baseline) converges the slowest among the four losses in 4D grids, and completely fails in 5D grids, while the other three losses remain robust in most of the cases. This is because the quadratic loss has the poorest exploration ability. Despite the differences in convergence speed, the $L_1$ errors between $P_T$ and $P_R$ are almost the same at convergence when using different regression functions.

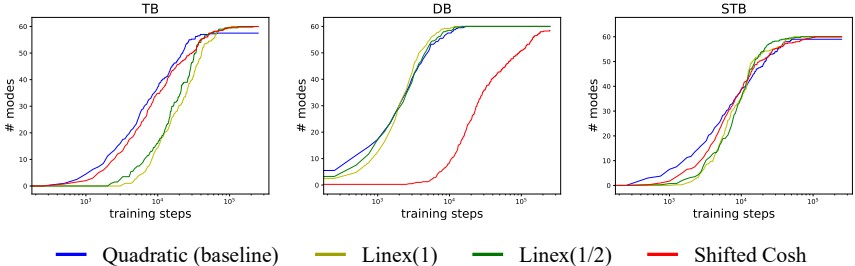

Figure 4: The number of modes found by the algorithm during training.

## 5.2 BIT-SEQUENCE GENERATION

In our second experimental setting, we study the bit-sequence generation task proposed by Malkin et al. (2022) and Tiapkin et al. (2024). The goal is to generate binary strings of length $n$ given a fixed word length $k \mid n$. In this setup, an $n$-bit string is represented as a sequence of $n/k$ $k$-bit words. The generation process starts with a sequence of $n/k$ special empty words. At each step, a valid action replaces an empty word with any $k$-bit word. Terminal states are sequences with no empty words. The reward is defined based on the minimal Hamming distance to any target mode in the given set $M \subset \mathbb{Z}_2^n$. Specifically, $R(x) = \exp\{-\min_{x' \in M} d(x, x')\}$.

In our experiments, we follow the setup in Tiapkin et al. (2024) where $n = 120, k = 8, |M| = 60$. $P_B$ is fixed to be uniform during training. We use the $\epsilon$-noisy forward policy with a random action probability of $0.001$ to sample training objects and the forward-looking style parameterizations for DB and STB experiments. We evaluate the number of modes found during training (the number of bit sequences in $M$ such that a candidate within a distance $\Delta = 30$ has been generated) as well as the Spearman Correlation between $P_T$ and $P_R$ over a test set, which has also been adopted by Malkin et al. (2022), Madan et al. (2023) and Tiapkin et al. (2024). Additional details can be found in Appendix E.2.

Table 3: The number of runs that find all modes within 250k steps, and the median of the steps before they find all modes.

|  | Quadratic (baseline) | Linex$(1)$ | Linex$(1/2)$ | Shifted-Cosh |
|---|---|---|---|---|
| TB | 1/5,  – | 5/5, 98.0k | 5/5, 111.2k | 4/5, **92.2k** |
| DB | 5/5, 13.4k | 5/5, **10.8k** | 5/5, 11.7k | 0/5,  – |
| STB | 4/5, 50.6k | 5/5, **20.3k** | 5/5, 55.9k | 5/5, 90.0k |

Table 4: The Spearman correlation between $P_T$ and $P_R$ over a test set (the higher the better). The failed runs that modal collapse happened are eliminated.

|  | Quadratic (baseline) | Linex$(1)$ | Linex$(1/2)$ | Shifted-Cosh |
|---|---|---|---|---|
| zero-forcing | ✓ | ✗ | ✗ | ✓ |
| TB | 0.8081($\pm$0.0159) | 0.7421($\pm$0.0216) | 0.7454($\pm$0.0021) | **0.8122**($\pm$0.0145) |
| DB | 0.7907($\pm$0.0175) | 0.7464($\pm$0.0107) | 0.7580($\pm$0.0132) | **0.8213**($\pm$0.0094) |
| STB | 0.8088($\pm$0.0169) | 0.7517($\pm$0.0246) | 0.7711($\pm$0.0190) | **0.8132**($\pm$0.0149) |

As shown in Figure 4, the quadratic loss seems to find new modes faster than the other three, but it always slows down and then is overtaken before finding all modes. As shown in Table 3, quadratic loss fails to find all modes in one of the five STB runs, and four out of the five TB runs. On the contrary, Linex(1) and Linex(1/2) succeed in finding all modes in all 15 runs with three different settings, and Linex(1) is always faster. The performance of shifted-Cosh varies from different algorithms. As we analyzed in Section 4.3, a zero-avoiding loss benefits exploration, while a zero-forcing loss does the opposite. These results are consistent with our analysis in general.

In this environment, the state space is so large that the training objects can not fully cover the whole space. Consequently, although the algorithms appear to converge, the distribution $\hat{P}_T$ only approximates $P_R$ rather than perfectly matching it. In such cases, zero-forcing losses have advantages

on the qualities of samples compared to non-zero-forcing ones. As shown in Table 4, zero-forcing losses (Quadratic and Shifted-Cosh) result in a higher correlation between $P_T$ and $P_R$, meaning that they fit the target distribution better within its support. Besides, we also observe that for TB GFlowNets with quadratic loss, the forward policy sometimes collapses to fitting only a small proportion of the modes in the target distribution, resulting in extremely low correlation. We eliminate these runs when presenting Table 4.

## 5.3 MOLECULE GENERATION

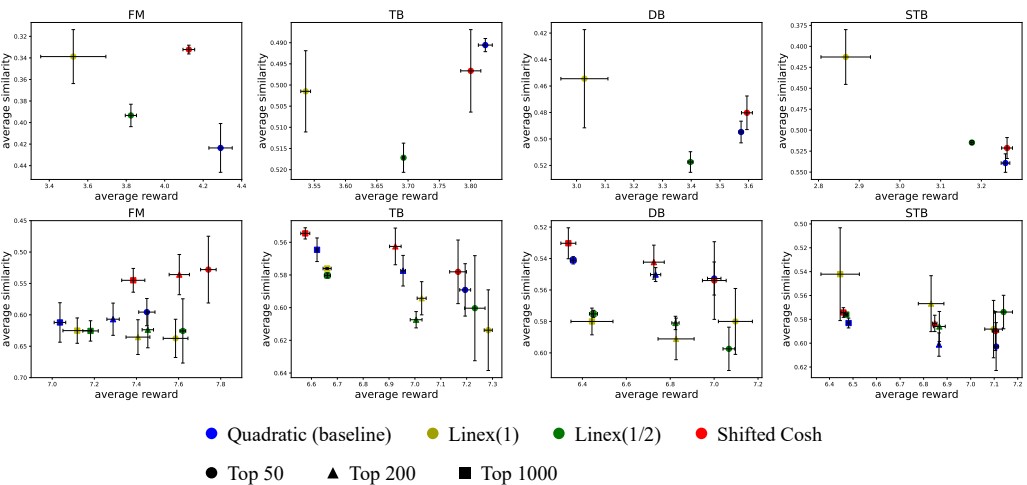

Figure 5: Molecule generation results. Top: Average reward and pair-wise similarities of all $200k$ generated molecules during each training episode. The similarities are calculated among a randomly chosen subset of 1000 molecules. Bottom: Average reward and pair-wise similarities of the top $k$ generated molecules during each training episode.

The goal of this task is to generate binders of the sEH (soluble epoxide hydrolase) protein by sequentially joining 'blocks' from a fixed library to the partial molecular graph (Jin et al. (2018)). The reward function is given by a pretrained proxy model given by Bengio et al. (2021), and then adjusted by a reward exponent hyperparameter $\beta$, i.e., $R(x) = \tilde{R}(x)^\beta$ where $\tilde{R}(x)$ is the output of the proxy model. For DB, TB, and STB experiments, the backward policies are fixed to be uniform. The training objects are sampled from the $\epsilon$-noisy forward policy with a random action probability of 0.05. Additional details can be found in Appendix E.3.

It can be seen in Figure 5 that zero-forcing objectives (Quadractic and shifted-Cosh) have a higher overall average reward, while zero-avoiding objectives (Linex(1) and Linex(1/2)) have lower overall similarities, meaning that the samples are more diverse. However, things become different when it comes to the top $k$ molecules, but Linex(1/2), which is neither zero-forcing nor zero-avoiding, demonstrates the best robustness among them.

## 6 CONCLUSION

In this work, we develop a principled and systematic approach for designing regression losses for efficient GFlowNets training. Specifically, we rigorously prove that distinct regression losses correspond to specific divergence measures, enabling us to design and analyze regression losses according to the desired properties of the corresponding divergence measures. Based on our theoretical framework, we designed three novel regression losses: Shifted-Cosh, Linex(1/2), and Linex(1). Through extensive evaluation across three benchmarks: hyper-grid, bit-sequence generation, and molecule generation, we show that our newly proposed losses are compatible with most existing training algorithms and significantly improve the performance of the algorithms in terms of convergence speed, sample diversity, and robustness.

ACKNOWLEDGEMENT

This work was supported by the National Natural Science Foundation of China Grants 52450016 and 52494974. The authors thank Professor Yang Yuan for discussions.

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

# Appendix

## A  UNIFYING TRAINING ALGORITHMS OF GFLOWNETS

An objective function for training GFlowNets is specified by five key components, the training objects $\mathcal{O}$, the parameterization mapping $\hat{p}_\theta$, the sampling and resampling weights $\mu$, the backward policy $P_B$ and the regression loss $g$. Most existing algorithms specify only one to two of the former four components.

### A.1  TRAINING OBJECTS AND PARAMETERIZATION MAPPING

The choice of these two components are usually coupled since the parameters are mapped to the flow functions defined on training objects. The choice of training objects include states, edges, partial trajectories and complete trajectories, corresponding to Flow-Matching GFlownets (FM-GFN, Bengio et al. 2021), Detailed-Balance GFlowNets (DB-GFN, Bengio et al. 2023), Sub-Trajectory-Balance GFlowNets (STB-GFN, Madan et al. 2023) and Trajectory-Balance GFlowNets (TB-GFN, Malkin et al. 2022), respectively. Detailed-Balance GFlowNets and Sub-Trajectory-Balance GFlowNets can be parameterized in different ways, the variants of which are Forward-Looking GFlowNets (FL-GFN, Pan et al. 2023a) and DAG GFlowNets (DAG-GFN, also called modified-DB or modified-STB, Deleu et al. 2022; Hu et al. 2024). These algorithms can be summarized in Table 5.

Table 5: The training objects and parameterization mappings of different GFlowNet training algorithms. Among the parameters, $\hat{P}_B$ can be either fixed or learned.

| Algorithm | Training Objects | Parameters | Parameterization mapping |
|---|---|---|---|
| FM | states | $\hat{F}(s \to s')$ | Equation (3), (4) |
| DB | transitions | $\hat{F}(s), \hat{P}_F(s'\|s)\,(, \hat{P}_B(s\|s'))$ | Equation (5), (6) |
| FL-DB | transitions | $\tilde{F}(s), \hat{P}_F(s'\|s)\,(, \hat{P}_B(s\|s'))$ | Equation (11), (12) |
| modified-DB | transitions | $\hat{P}_F(s'\|s)\,(, \hat{P}_B(s\|s'))$ | Equation (15), (16) |
| TB | complete trajectories | $\hat{Z}, \hat{P}_F(s'\|s)\,(, \hat{P}_B(s\|s'))$ | Equation (7), (8) |
| STB | partial trajectories | $\hat{F}(s), \hat{P}_F(s'\|s)\,(, \hat{P}_B(s\|s'))$ | Equation (9), (10) |
| FL-STB | partial trajectories | $\tilde{F}(s), \hat{P}_F(s'\|s)\,(, \hat{P}_B(s\|s'))$ | Equation (13), (14) |
| modified-STB | partial trajectories | $\hat{P}_F(s'\|s)\,(, \hat{P}_B(s\|s'))$ | Equation (17), (18) |

**Flow-Matching GFlowNets (FM-GFN).**   An FM-GFN is parameterized by an edge-flow function $\hat{F} : E \to \mathbb{R}_+$. It uniquely determines a valid flow network if and only if the **flow-matching conditions** hold:

$$\forall s \in V \setminus \{s_o, s_f\}, \quad \sum_{(s' \to s) \in E} \hat{F}(s' \to s) = R(s) + \sum_{\substack{(s \to s'') \in E \\ s'' \neq s_f}} \hat{F}(s \to s'')$$

The **flow-matching loss** for state $s$ is defined as

$$L_{FM}(s) = \frac{1}{2} \left( \log \frac{\hat{p}_B(s)}{\hat{p}_F(s)} \right)^2$$

$$\text{where } \hat{p}_F(s) = \sum_{(s' \to s) \in E} \hat{F}(s' \to s) \tag{3}$$

$$\hat{p}_B(s) = R(s) + \sum_{\substack{(s \to s'') \in E \\ s'' \neq s_f}} \hat{F}(s \to s'') \tag{4}$$

**Detailed-Balance GFlowNets (DB-GFN).**   A DB-GFN is parameterized by a state-flow function $\hat{F} : V \setminus \{s_f\} \to \mathbb{R}_+$, a forward probability function $\hat{P}_F : V \setminus \{s_f\} \to \Delta(V)$ and a backward probability function $\hat{P}_B : V \setminus \{s_0, s_f\} \to \Delta(V)$. They uniquely determine a valid flow network if

and only if the **detailed balance conditions** hold:

$$\forall s \in S^f, \hat{F}(s)\hat{P}_F(s_f|s) = R(s)$$

$$\forall (s \to s') \in E, s' \neq s_f, \hat{F}(s)\hat{P}_F(s'|s) = \hat{F}(s')\hat{P}_B(s|s')$$

The **detailed-balance loss** for transition $s \to s'$ is defined as

$$L_{DB}(s \to s') = \frac{1}{2}\left(\log\frac{\hat{p}_B(s \to s')}{\hat{p}_F(s \to s')}\right)^2$$

$$\text{where } \hat{p}_F(s \to s') = \hat{F}(s)\hat{P}_F(s'|s) \tag{5}$$

$$\hat{p}_B(s \to s') = \begin{cases} \hat{F}(s')\hat{P}_B(s|s') & , s' \neq s_f \\ R(s) & , s' = s_f \end{cases} \tag{6}$$

**Trajectory-Balance GFlowNets (TB-GFN).** A TB-GFN is parameterized by a total flow function $\hat{Z}$, a forward probability function $\hat{P}_F : V \setminus \{s_f\} \to \Delta(V \setminus \{s_0\})$ and a backward probability function $\hat{P}_B : V \setminus \{s_0\} \to \Delta(V \setminus \{s_f\})$. They uniquely determine a GFlowNet if and only if the **trajectory balance conditions** hold:

$$\forall \tau = (s_0 = s_o, s_1, \cdots, s_{T-1}, s_T = s_f), \hat{Z}\prod_{t=0}^{T=1}\hat{P}_F(s_{t+1}|s_t) = R(s_{T-1})\prod_{t=1}^{T-1}\hat{P}_B(s_{t-1}|s_t)$$

The **trajectory-balance loss** for complete trajectory $\tau = (s_0 = s_o, s_1, \cdots, s_{T-1}, s_T = s_f)$ is defined as

$$L_{TB}(\tau) = \frac{1}{2}\left(\log\frac{\hat{p}_B(\tau)}{\hat{p}_F(\tau)}\right)^2$$

$$\text{where } \hat{p}_F(\tau) = \hat{Z}\prod_{t=0}^{T=1}\hat{P}_F(s_{t+1}|s_t) \tag{7}$$

$$\hat{p}_B(\tau) = R(s_{T-1})\prod_{t=1}^{T-1}\hat{P}_B(s_{t-1}|s_t) \tag{8}$$

**Sub-Trajectory-Balance GFlowNets (STB-GFN).** An STB-GFN uses the same parameters as a DB-GFN with an alternative loss, the **sub-trajectory-balance loss**. It is defined for partial trajectory $\iota = (s_0, s_1, \cdots, s_{T-1}, s_T)$ as

$$L_{STB}(\iota) = \frac{1}{2}\left(\log\frac{\hat{p}_B(\iota)}{\hat{p}_F(\iota)}\right)^2$$

$$\text{where } \hat{p}_F(\iota) = \hat{F}(s_0)\prod_{t=0}^{T=1}\hat{P}_F(s_{t+1}|s_t) \tag{9}$$

$$\hat{p}_B(\iota) = \begin{cases} \hat{F}(s_T)\prod_{t=1}^{T}\hat{P}_B(s_{t-1}|s_t) & , s_T \neq s_f \\ R(s_{T-1})\prod_{t=1}^{T-1}\hat{P}_B(s_{t-1}|s_t) & , s_T = s_f \end{cases} \tag{10}$$

**Forward-looking GFlowNets (FL-GFN).** FL-GFNs require the assumption that the reward function can be extended to the whole state space, instead of restricted to only terminal states. The parameters of FL-GFN are quite similar to that of the original DB GFlowNets and STB GFlowNets, including a forward-looking state-flow function $\tilde{F} : V \setminus \{s_f\} \to \mathbb{R}_+$, a forward probability function $\hat{P}_F : V \setminus \{s_f\} \to \Delta(V)$ and a backward probability function $\hat{P}_B : V \setminus \{s_0, s_f\} \to \Delta(V)$. The only difference is that the original state-flow function $\hat{F}$ is replaced by the forward-looking version $\tilde{F}$, following $\hat{F}(s) = R(s)\tilde{F}(s)$. The **forward-looking detailed-balance loss** and **forward-looking**

**sub-trajectory-balance loss** can be obtained by substituting them with the original ones:

$$L_{\text{FL-DB}}(s \to s') = \frac{1}{2}\left(\log \frac{\hat{p}_B(s \to s')}{\hat{p}_F(s \to s')}\right)^2$$

$$\text{where } \hat{p}_F(s \to s') = R(s)\tilde{F}(s)\hat{P}_F(s'|s) \tag{11}$$

$$\hat{p}_B(s \to s') = \begin{cases} R(s')\tilde{F}(s')\hat{P}_B(s|s'), & s' \neq s_f \\ R(s), & s' = s_f \end{cases} \tag{12}$$

$$L_{\text{FL-STB}}(\iota) = \frac{1}{2}\left(\log \frac{\hat{p}_B(\iota)}{\hat{p}_F(\iota)}\right)^2$$

$$\text{where } \hat{p}_F(\iota) = R(s_0)\tilde{F}(s_0)\prod_{t=0}^{T=1}\hat{P}_F(s_{t+1}|s_t) \tag{13}$$

$$\hat{p}_B(\iota) = \begin{cases} R(s_T)\tilde{F}(s_T)\prod_{t=1}^{T}\hat{P}_B(s_{t-1}|s_t), & s_T \neq s_f \\ R(s_{T-1})\prod_{t=1}^{T-1}\hat{P}_B(s_{t-1}|s_t), & s_T = s_f \end{cases} \tag{14}$$

**DAG GFlowNets (DAG-GFN).** DAG-GFNs require that each state is terminated and has a non-zero reward. Then according to the detailed-balance condition, $\hat{F}(s) = \frac{R(s)}{\hat{p}_F(s_f|s)}$ for all $s$. Therefore, the flow network can be parameterized by only the forward probability function $\hat{P}_F : V \setminus \{s_f\} \to \Delta(V)$ and the backward probability function $\hat{P}_B : V \setminus \{s_0, s_f\} \to \Delta(V)$. The **modified detailed-balance loss** and **modified sub-trajectory-balance loss** can be obtained by substituting them into the original ones:

$$L_{\text{modified-DB}}(s \to s') = \frac{1}{2}\left(\log \frac{\hat{p}_B(s \to s')}{\hat{p}_F(s \to s')}\right)^2$$

$$\text{where } \hat{p}_F(s \to s') = \frac{R(s)\hat{P}_F(s'|s)}{\hat{P}_F(s_f|s)} \tag{15}$$

$$\hat{p}_B(s \to s') = \begin{cases} \frac{R(s')\hat{P}_B(s|s')}{\hat{P}_F(s_f|s')} & , s' \neq s_f \\ R(s) & , s' = s_f \end{cases} \tag{16}$$

$$L_{\text{modified-STB}}(\iota) = \frac{1}{2}\left(\log \frac{\hat{p}_B(\iota)}{\hat{p}_F(\iota)}\right)^2$$

$$\text{where } \hat{p}_F(\iota) = \frac{R(s_0)}{\hat{P}_F(s_f|s_0)}\prod_{t=0}^{T=1}\hat{P}_F(s_{t+1}|s_t) \tag{17}$$

$$\hat{p}_B(\iota) = \begin{cases} \frac{R(s_T)}{\hat{P}_F(s_f|s_T)}\prod_{t=1}^{T}\hat{P}_B(s_{t-1}|s_t) & , s_T \neq s_f \\ R(s_{T-1})\prod_{t=1}^{T-1}\hat{P}_B(s_{t-1}|s_t) & , s_T = s_f \end{cases} \tag{18}$$

## A.2 SAMPLING AND RESAMPLING WEIGHTS

There exist various strategies to sample training objects to enhance exploration and hence accelerate convergence. The usual practice is to use the forward policy, the backward policy, a tempered or $\epsilon$-noisy version of them, an offline dataset, or a mixture of these strategies. Other choices include using a reward prioritized replay buffer (Shen et al., 2023), applying Thompson sampling (Rector-Brooks et al., 2023), local search (Kim et al., 2024d) or genetic search (Kim et al., 2024a) to the sampled trajectories for extra samples, increasing greediness according to state-action value $Q$ (Lau et al., 2024), etc. The sampled objects may also be reweighed. For example, STB-GFN weights each partial trajectory by a factor proportional to $\lambda^l$, where $l$ is its length and $\lambda$ is a hyper-parameter.

## A.3 BACKWARD POLICY

The most common choice of $P_B$ is to either fix it to be uniform or simultaneously train it using the same objective as the forward policy. Other criteria include matching a (possibly non-Markovian)

prior (Shen et al., 2023), maximizing the entropy of the corresponding forward policy (Mohammad-pour et al., 2024) and learning a pessimistic one that focuses on observed trajectories (Jang et al., 2024a).

## B   THEOREM 4.1 AND ITS PROOF

**Theorem B.1** (An extension of Theorem 4.1). *Let $\theta_F$ and $\theta_B$ be the parameters for forward and backward policies, respectively. For each minimal cut $C \in \mathcal{C}$, the restrictions of both forward and backward flow functions on $C$ can be viewed as unnormalized distributions over it, denoted as $\hat{p}_F^C$ and $\hat{p}_B^C$, respectively.*

*If there exists $w : \mathcal{C} \to \mathbb{R}_+$ such that $\mu(o) = \hat{p}_F(o) \sum_{C \in \mathcal{C}, o \in C} w(C)$ for any $o \in \mathcal{O}$, then*

$$\nabla_{\theta_F} \mathcal{L}_{\mathcal{O}, \hat{p}_\theta, \mu, P_B, g} = \nabla_{\theta_F} \sum_{C \in \mathcal{C}} w(C) D_{f_1}(\hat{p}_B^C \| \hat{p}_F^C), \text{where } f_1(t) = t \int_1^t \frac{g'(\log s)}{s^2} ds$$

$$\nabla_{\theta_B} \mathcal{L}_{\mathcal{O}, \hat{p}_\theta, \mu, P_B, g} = \nabla_{\theta_B} \sum_{C \in \mathcal{C}} w(C) D_{f_2}(\hat{p}_B^C \| \hat{p}_F^C), \text{where } f_2(t) = g(\log t)$$

*If there exists $w : \mathcal{C} \to \mathbb{R}_+$ such that $\mu(o) = \hat{p}_B(o) \sum_{C \in \mathcal{C}, o \in C} w(C)$ for any $o \in \mathcal{O}$, then*

$$\nabla_{\theta_F} \mathcal{L}_{\mathcal{O}, \hat{p}_\theta, \mu, P_B, g} = \nabla_{\theta_F} \sum_{C \in \mathcal{C}} w(C) D_{f_3}(\hat{p}_B^C \| \hat{p}_F^C), \text{where } f_3(t) = tg(\log t)$$

$$\nabla_{\theta_B} \mathcal{L}_{\mathcal{O}, \hat{p}_\theta, \mu, P_B, g} = \nabla_{\theta_B} \sum_{C \in \mathcal{C}} w(C) D_{f_4}(\hat{p}_B^C \| \hat{p}_F^C), \text{where } f_4(t) = \int_1^t g'(\log s) ds$$

*Proof.* We prove the theorem by deriving the correspondence. Specifically, assume $\mu(o) = \hat{p}_F(o) \sum_{C \in \mathcal{C}, o \in C} w(C)$. Then,

$$\nabla_{\theta_F} \sum_{C \in \mathcal{C}} w(C) D_{f_1}(\hat{p}_B^C \| \hat{p}_F^C) = \sum_{C \in \mathcal{C}} w(C) \sum_{o \in C} \nabla_{\theta_F} \left[ \hat{p}_F^C(o) f_1 \left( \frac{\hat{p}_B^C(o)}{\hat{p}_F^C(o)} \right) \right]$$

$$= \sum_{C \in \mathcal{C}} w(C) \sum_{o \in C} \left[ f_1 \left( \frac{\hat{p}_B^C(o)}{\hat{p}_F^C(o)} \right) - \frac{\hat{p}_B^C(o)}{\hat{p}_F^C(o)} f_1' \left( \frac{\hat{p}_B^C(o)}{\hat{p}_F^C(o)} \right) \right] \nabla_{\theta_F} \hat{p}_F^C(o)$$

$$= \sum_{C \in \mathcal{C}} w(C) \sum_{o \in C} -g' \left( \log \frac{\hat{p}_B^C(o)}{\hat{p}_F^C(o)} \right) \nabla_{\theta_F} \hat{p}_F^C(o)$$

$$= \sum_{C \in \mathcal{C}} w(C) \sum_{o \in C} \hat{p}_F^C(o) g' \left( \log \frac{\hat{p}_B^C(o)}{\hat{p}_F^C(o)} \right) \left( -\frac{1}{\hat{p}_F^C(o)} \right) \nabla_{\theta_F} \hat{p}_F^C(o)$$

$$= \sum_{o \in \mathcal{O}} \mu(o) \nabla_{\theta_F} g \left( \log \frac{\hat{p}_B^C(o)}{\hat{p}_F^C(o)} \right)$$

$$= \nabla_{\theta_F} \mathcal{L}_{\mathcal{O}, \hat{p}_\theta, \mu, P_B, g}$$

$$\nabla_{\theta_B} \sum_{C \in \mathcal{C}} w(C) D_{f_2}(\hat{p}_B^C \| \hat{p}_F^C) = \sum_{C \in \mathcal{C}} w(C) \sum_{o \in C} \nabla_{\theta_B} \left[ \hat{p}_F^C(o) f_2 \left( \frac{\hat{p}_B^C(o)}{\hat{p}_F^C(o)} \right) \right]$$

$$= \sum_{C \in \mathcal{C}} w(C) \sum_{o \in C} \left[ \hat{p}_F^C(o) \nabla_{\theta_B} g \left( \log \frac{\hat{p}_B^C(o)}{\hat{p}_F^C(o)} \right) \right]$$

$$= \sum_{o \in \mathcal{O}} \mu(o) \nabla_{\theta_B} g \left( \log \frac{\hat{p}_B^C(o)}{\hat{p}_F^C(o)} \right)$$

$$= \nabla_{\theta_B} \mathcal{L}_{\mathcal{O}, \hat{p}_\theta, \mu, P_B, g}$$

In the second case, suppose $\mu(o) = \hat{p}_B(o) \sum_{C \in \mathcal{C}, o \in C} w(C)$. Then,

$$\nabla_{\theta_F} \sum_{C \in \mathcal{C}} w(C) D_{f_3}(\hat{p}_B^C \| \hat{p}_F^C) = \sum_{C \in \mathcal{C}} w(C) \sum_{o \in C} \nabla_{\theta_F} \left[ \hat{p}_F^C(o) f_3 \left( \frac{\hat{p}_B^C(o)}{\hat{p}_F^C(o)} \right) \right]$$

$$= \sum_{C \in \mathcal{C}} w(C) \sum_{o \in C} \left[ f_3 \left( \frac{\hat{p}_B^C(o)}{\hat{p}_F^C(o)} \right) - \frac{\hat{p}_B^C(o)}{\hat{p}_F^C(o)} f_3' \left( \frac{\hat{p}_B^C(o)}{\hat{p}_F^C(o)} \right) \right] \nabla_{\theta_F} \hat{p}_F^C(o)$$

$$= \sum_{C \in \mathcal{C}} w(C) \sum_{o \in C} -\frac{\hat{p}_B^C(o)}{\hat{p}_F^C(o)} g' \left( \log \frac{\hat{p}_B^C(o)}{\hat{p}_F^C(o)} \right) \nabla_{\theta_F} \hat{p}_F^C(o)$$

$$= \sum_{C \in \mathcal{C}} w(C) \sum_{o \in C} \hat{p}_B^C(o) g' \left( \log \frac{\hat{p}_B^C(o)}{\hat{p}_F^C(o)} \right) \left( -\frac{1}{\hat{p}_F^C(o)} \right) \nabla_{\theta_F} \hat{p}_F^C(o)$$

$$= \sum_{o \in \mathcal{O}} \mu(o) \nabla_{\theta_F} g \left( \log \frac{\hat{p}_B^C(o)}{\hat{p}_F^C(o)} \right)$$

$$= \nabla_{\theta_F} \mathcal{L}_{\mathcal{O}, \hat{p}_\theta, \mu, P_B, g}$$

$$\nabla_{\theta_B} \sum_{C \in \mathcal{C}} w(C) D_{f_4}(\hat{p}_B^C \| \hat{p}_F^C) = \sum_{C \in \mathcal{C}} w(C) \sum_{o \in C} \nabla_{\theta_B} \left[ \hat{p}_F^C(o) f_4 \left( \frac{\hat{p}_B^C(o)}{\hat{p}_F^C(o)} \right) \right]$$

$$= \sum_{C \in \mathcal{C}} w(C) \sum_{o \in C} f_4' \left( \frac{\hat{p}_B^C(o)}{\hat{p}_F^C(o)} \right) \nabla_{\theta_B} \hat{p}_B^C(o)$$

$$= \sum_{C \in \mathcal{C}} w(C) \sum_{o \in C} \hat{p}_B^C(o) g' \left( \log \frac{\hat{p}_B^C(o)}{\hat{p}_F^C(o)} \right) \frac{1}{\hat{p}_B^C(o)} \nabla_{\theta_B} \hat{p}_B^C(o)$$

$$= \sum_{o \in \mathcal{O}} \mu(o) \nabla_{\theta_B} g \left( \log \frac{\hat{p}_B^C(o)}{\hat{p}_F^C(o)} \right)$$

$$= \nabla_{\theta_B} \mathcal{L}_{\mathcal{O}, \hat{p}_\theta, \mu, P_B, g}$$

$\square$

## C  INTERPRETATION OF THEOREM 4.1 FOR DIFFERENT KINDS OF LOSSES

### C.1  FLOW MATCHING LOSS

For any $s \in V$, let $l(s)$ be the length of the longest trajectory from $s_o$ to $s$. For any $(s \to s') \in E$, if $l(s) + 1 < l(s')$, then we insert $l(s') - l(s) - 1$ virtual states on this edge, denoted as $s_{(s \to s'), l}$ for $l(s) < l < l(s')$, and define

$$\hat{p}_F(s_{(s \to s'), l}) = \hat{p}_B(s_{(s \to s'), l}) = \hat{F}(s \to s')$$

then these virtual states have no contribution to the total loss, thus we can assign to them arbitrary weights.

Let $V^i$ be the collections of states in layer $i$, and let $w(V^i) = 1$, then we have

$$\mu(s) = \hat{p}_F^{V^{l(s)}}(s) = \hat{p}_F(s)$$

### C.2  DETAILED BALANCE LOSS

For any $s \in V$, let $l(s)$ be the length of the longest trajectory from $s_o$ to $s$. For any $(s \to s') \in E$, if $l(s) + 1 < l(s')$, then we insert $l(s') - l(s) - 1$ virtual states on this edge, denoted as $s_{(s \to s'), l}$ for $l(s) < l < l(s')$, and define

$$\hat{p}_F^l(s \to s') = \hat{p}_F(s \to s')$$

$$\hat{p}_B^l(s \to s') = \begin{cases} \hat{p}_F(s \to s') & , l < l(s') \\ \hat{p}_B(s \to s') & , l = l(s') \end{cases}$$

then these virtual transitions have no contribution to the total loss, thus we can assign to them arbitrary weights.

Let $E^i$ be the collections of edges from layer $i$ to layer $i+1$, and let $w(E^i) = 1$, then we have

$$\mu(s \to s') = \hat{p}_F^{E^{l(s)}}(s \to s') = \hat{p}_F(s \to s')$$

### C.3 Sub-Trajectory Balance Loss

Assume that $G$ is a graded DAG with $L+1$ layers. Suppose $\tau = (s_0 = s_o, s_1, \cdots, s_L = s_f)$ is a complete trajectory, we use $\tau_{i:j} = (s_i, s_{i+1}, \cdots, s_j)$ to denote a partial trajectory. Let $\mathcal{T}^{i:j}$ be the collections of trajectories from layer $i$ to layer $j$, then

$$\mu(\iota) = \sum_{\tau:\iota=\tau_{i:j}} \hat{P}_F(\tau) \frac{\lambda^{j-i}}{\sum_{0 \leq i < j \leq L} \lambda^{j-i}}$$

$$\approx \frac{\lambda^{j-i}}{\sum_{0 \leq i < j \leq L} \lambda^{j-i}} \hat{p}_F^{\mathcal{T}^{i:j}}(\iota)$$

Hence $w(\mathcal{T}^{i:j}) = \frac{\lambda^{j-i}}{\sum_{0 \leq i < j \leq L}}$ and 0 otherwise.

## D Proof of Theorem 4.7

**Theorem D.1.** *Let $\mathcal{L}$ be an objective function for training GFlowNets, whose regression loss $g$ corresponds to $D_f$ according to Theorem 4.1. If $D_f$ is zero-forcing, then $\mathcal{L}$ and $g$ are both zero-forcing. If $D_f$ is zero-avoiding, then $\mathcal{L}$ and $g$ are both zero-avoiding.*

*Proof.* Assume that $D_f$ is zero-forcing, and $\hat{P}_T(s; \theta^*) > 0$ for some terminating state $s$. Then there exists a trajectory $\tau = (s_o, \cdots, s, s_f)$ such that $\hat{P}_F(\tau; \theta) > 0$, thus

$$\hat{p}_F^C(o) = \hat{p}_F(o) > 0$$

for any $o \in \tau, o \in C, w(C) > 0$. Since $D_f$ is zero-forcing, $\hat{p}_B(o) = \hat{p}_B^C(o) > 0$ for any $o \in \tau$, meaning that $\hat{P}_B(\tau) > 0$ and $R(s) > 0$. Thus, $R(s) = 0$ implies $\hat{P}_T(s; \theta) = 0$, so $\mathcal{L}$ is zero-forcing, and then $g$ is zero-forcing as well.

Similarly, assume that $D_f$ is zero-avoiding, and $R(s) > 0$ for some terminating state $s$. Then there exists a trajectory $\tau = (s_o, \cdots, s, s_f)$ such that $\hat{P}_B(\tau) > 0$, thus

$$\hat{p}_B^C(o) = \hat{p}_B(o) > 0$$

for any $o \in \tau, o \in C, w(C) > 0$. Since $D_f$ is zero-avoiding, $\hat{p}_F(o) = \hat{p}_F^C(o) > 0$ for any $o \in \tau$, meaning that $\hat{P}_F(\tau; \theta) > 0$, so $\hat{P}_T(s; \theta) > 0$. Thus, $R(s) > 0$ implies that $\hat{P}_T(s; \theta) > 0$, so $\mathcal{L}$ is zero-avoiding, and then $g$ is zero-avoiding as well. □

## E Experimental Details

### E.1 Hyper-grid

Our implementation of the baselines is based on Tiapkin et al. (2024). All models are parameterized by an MLP with 2 hidden layers of 256 neurons. We train the model with Adam optimizer using a batch size of 16 and a learning rate of 0.001. For the TB case, we use a larger learning rate of 0.1 for learnable total flow $\hat{Z}$. For STB parameter $\lambda$, we use the value of 0.9 following Tiapkin et al. (2024) and Madan et al. (2023). We repeat each experiment 3 times using different random seeds. In each run, we train the models until 800k trajectories have been collected, and the empirical sample distribution is computed over the last 80k seen trajectories.

### E.2 BIT-SEQUENCE GENERATION

In this experiment, our implementation of the baselines is based on Tiapkin et al. (2024) and Pan et al. (2023a). The model is a 3-layer Transformer with 64 hidden units and 8 attention heads per layer. We train the model with Adam optimizer using a batch size of 16 and a learning rate of 0.001. For the TB case, we use a larger learning rate of 0.002 for learnable total flow $\hat{Z}$. For STB parameter $\lambda$, we use the value of 1.5. Following Tiapkin et al. (2024), we use a reward exponent of 2. To calculate the Spearman Correlation, we use the same Monte-Carlo estimation for $P_T$ as Zhang et al. (2022) and Tiapkin et al. (2024), namely

$$P_T(x) \approx \frac{1}{N} \sum_{i=1}^{N} \frac{P_F(\tau^i)}{P_B(\tau^i|x)}$$

with $N = 10$. We repeat each experiment 5 times using different random seeds.

### E.3 MOLECULE GENERATION

In the molecule generation experiment, our implementation of the baselines is based on Tiapkin et al. (2024). We use Message Passing Neural Networks (MPNN) as the model architecture. We train the model with Adam optimizer using a batch size of 4 and a learning rate of 0.0005. We use a reward exponent of 4, and the STB parameter $\lambda$ is set to 0.99. We repeat each experiment 4 times using different random seeds. In each run, We train the models for 50000 steps, generating 200k molecules.

## F MORE DIVERGENCE-BASED LOSSES

Apart from the four representative divergence-based losses in Section 4.3, we also derive another five novel losses from some well-known divergence measures, including the forward and backward $\chi^2$ distance, total variation distance, symmetric KL divergence and Jensen-Shannon divergence (See Table 6).

Table 6: Five well-known $f$-divergences and their corresponding regression losses.

| Divergence | $f(t)$ | $g(t)$ | Zero-forcing | Zero-avoiding |
|---|---|---|---|---|
| Forward $\chi^2$ | $\frac{1}{2}(t-1)^2$ | $\frac{1}{4}\left(e^{2t} - 2t - 1\right)$ | | ✓ |
| Reverse $\chi^2$ | $\frac{1}{2}\left(t + \frac{1}{t} - 2\right)$ | $e^{-t} + t - 1$ | ✓ | |
| Total Variation | $\frac{1}{2}|t-1|$ | $\frac{1}{2}|t|$ | | |
| Symmetric KL | $\frac{1}{2}(t-1)\log t$ | $\frac{1}{2}\left(e^t + \frac{1}{2}t^2 - t - 1\right)$ | ✓ | ✓ |
| JS Divergence | $\frac{1}{2}\left(t\log t - (t+1)\log\left(\frac{t+1}{2}\right)\right)$ | $\frac{1}{2}\int_0^t \log\left(\frac{1+e^x}{2}\right) dx$ | | |

$\chi^2$ **distance** . The $\chi^2$ distance between $p$ and $q$ is defined as

$$\chi^2(p||q) = \frac{1}{2} \sum_{x \in \mathcal{X}} \frac{(p(x) - q(x))^2}{q(x)} \tag{19}$$

It is a special case of $\alpha$-divergence with $\alpha = 2$, or $f$-divergence with $f(t) = \frac{1}{2}(t-1)^2$. According to **Theorem** 4.1, we obtain the corresponding regression loss Linex(2): $g(t) = \frac{1}{4}\left(e^{2t} - 2t - 1\right)$.

By exchanging $p$ and $q$ in (19), we obtain the reverse $\chi^2$ distance, which is a special case of $\alpha$-divergence with $\alpha = -1$, or $f$-divergence with $f(t) = \frac{1}{2}\left(t + \frac{1}{t} - 2\right)$. The corresponding regression loss Linex(-1): $g(t) = e^{-t} + t - 1$.

The forward $\chi^2$ distance and Linex(2) are zero-avoiding, while the reverse $\chi^2$ and Linex(-1) distance are zero-forcing.

**Total Variation.** The total variation between $p$ and $q$ is defined as

$$TV(p||q) = \frac{1}{2} \sum_{x \in \mathcal{X}} |p(x) - q(x)|$$

It corresponds to the $f$-divergence with $f(t) = \frac{1}{2}|t-1|$, and the regression loss $g(t) = \frac{1}{2}|t|$. Since $f(0) = f'(\infty) = \frac{1}{2}$, this loss function is neither zero-forcing nor zero-avoiding.

**Symmetric KL Divergence.** The symmetric KL divergence between $p$ and $q$ is defined as

$$
\begin{aligned}
D_{sKL}(p||q) =& \frac{1}{2}\left(D_{KL}(p||q) + D_{KL}(q||p)\right) \\
=& \frac{1}{2}\sum_{x\in\mathcal{X}}\left(p(x)\log\frac{p(x)}{q(x)} + q(x)\log\frac{q(x)}{p(x)}\right)
\end{aligned}
$$

It corresponds to the $f$-divergence with $f(t) = \frac{1}{2}(t-1)\log t$, and the regression loss $g(t) = \frac{1}{2}\left(e^t + \frac{1}{2}t^2 - t - 1\right)$. Since $f(0) = f'(\infty) = \infty$, this loss function is both zero-forcing and zero-avoiding.

**Jensen-Shannon Divergence.** The Jensen-Shannon divergence (JS divergence) between $p$ and $q$ is defined as

$$
\begin{aligned}
D_{JS}(p||q) =& \frac{1}{2}\left(D_{KL}(p||\frac{p+q}{2}) + D_{KL}(q||\frac{p+q}{2})\right) \\
=& \frac{1}{2}\sum_{x\in\mathcal{X}}\left(p(x)\log\frac{2p(x)}{p(x)+q(x)} + q(x)\log\frac{2q(x)}{p(x)+q(x)}\right)
\end{aligned}
$$

It corresponds to the $f$-divergence with $f(t) = \frac{1}{2}\left(t\log t - (t+1)\log\left(\frac{t+1}{2}\right)\right)$, and the regression loss $g(t) = \frac{1}{2}\int_0^t \log\left(\frac{1+e^x}{2}\right)dx$. Since $f(0) = f'(\infty) = \frac{1}{2}\log 2$, this loss function is neither zero-forcing nor zero-avoiding.

We evaluate their performance on bit-sequence generation task using the same metrics (Please refer to Section 5.2 and Appendix E.2 for details). It turns out that losses with the same zero-forcing or zero-avoiding properties lead to similar behaviors.

Table 7: The number of runs that find all modes within 250k steps, and the median of the steps before they find all modes.

|  | TB | DB | STB |
|---|---|---|---|
| Reverse KL (baseline) | 1/5,  – | 5/5, 13.4k | 4/5, 50.6k |
| Reverse $\chi^2$ | 0/5,  – | 0/5,  – | 0/5,  – |
| Forward KL | 5/5, 98.0k | 5/5, 10.8k | 5/5, 20.3k |
| Forward $\chi^2$ | 5/5, **80.3k** | 5/5, **8.1k** | 5/5, **10.2k** |
| Hellinger | 5/5, 111.2k | 5/5, 11.7k | 5/5, 55.9k |
| Total Variation | 1/5,  – | 5/5, 47.1k | 2/5,  – |
| Jensen-Shannon | 4/5, 162.2k | 5/5, 12.8k | 3/5, 165.2k |
| Shifted-Cosh | 4/5, 92.2k | 0/5,  – | 5/5, 90.0k |
| Symmetric KL | 4/5, 122.2k | 5/5, 13.7k | 5/5, 27.5k |

Table 8: The Spearman correlation between $P_T$ and $P_R$ over a test set (the higher the better). The failed runs where modal collapse happened are eliminated.

|  | TB | DB | STB |
| --- | --- | --- | --- |
| Reverse KL (baseline) | $\underline{0.8081}(\pm0.0159)$ | $0.7907(\pm0.0175)$ | $\underline{0.8088}(\pm0.0169)$ |
| Reverse | $\underline{0.8074}(\pm0.0129)$ | – | $\underline{0.7899}(\pm0.0166)$ |
| Forward KL | $0.7421(\pm0.0216)$ | $0.7464(\pm0.0107)$ | $0.7517(\pm0.0246)$ |
| Forward $\chi^2$ | $0.7507(\pm0.0174)$ | $0.7266(\pm0.0178)$ | $0.7439(\pm0.0126)$ |
| Hellinger | $0.7454(\pm0.0021)$ | $0.7580(\pm0.0132)$ | $0.7711(\pm0.0190)$ |
| Total Variation | $\underline{0.7893}(\pm0.0144)$ | $0.7266(\pm0.0178)$ | – |
| Jensen-Shannon | $\underline{0.7852}(\pm0.0256)$ | $0.7542(\pm0.0046)$ | $0.7640(\pm0.0213)$ |
| Shifted-Cosh | $\mathbf{0.8122}(\pm0.0145)$ | $\mathbf{0.8213}(\pm0.0094)$ | $\mathbf{0.8132}(\pm0.0149)$ |
| Symmetric KL | $\underline{0.7908}(\pm0.0235)$ | $0.7630(\pm0.0097)$ | $\underline{0.7886}(\pm0.0227)$ |

