# OpenReview forum: "Beyond Squared Error: Exploring Loss Design for Enhanced Training of Generative Flow Networks"
_ICLR.cc/2025/Conference — ICLR 2025 Spotlight_

### Official Review · Reviewer_8npT · 2024-11-02

**Soundness:** 4
**Presentation:** 4
**Contribution:** 3
**Rating:** 8
**Confidence:** 3

**Summary:**

The authors propose to modify the loss function of Gflownet (which has been completely overlooked by prior work). They show that dinstinct losses lead to different divergences. They propose three new loss functions, evaluate them extensively on diverse benchmarks.

**Strengths:**

Contributions:
- generalizing the objective function of gflownet
- derive impact of loss function on the gradient
- define zero-forcing (encourage epxloitation) and zero-avoiding (encourage exploration) as two key properties induced by certain loss functions
- They create 3 new losses (alonside the existing quadratic loss) to tackle all 4 possible combinations (with/without zero-avoiding, with/without zero-forcing)
- Linex(1) corresponds to the KL divergence
- experiments on 3 datasets
	- Non-zero-forcing losses (Linex(1) and Linex(0.5)) converge faster on hyper-grid
	- Linex(1) obtains all the modes almost always the fastest, but spearman corr between train and test is highest for shifted-cos on bit-sequence
	- Linex(1) tends increase diversity while quadratic and shifted-cos give higher quality (high average rewards) on molecule generation

Paper is well written.

**Weaknesses:**

- it would be nice to have a few more choices of losses, taking inspiration let say from f-divergences (Reverse-KL, JSD, etc.)
- there may be more than just zero-forcing and zero-avoiding to the key properties of loss functions hence why studying more losses would be helpful
- it would be nice to let say consider hybrid methods with some kind of annealing. For example, why not use Linex(1) for its fast convergence to a large number of nodes, before then transitioning to shifted-cos for higher rewards around those now-discovered modes.

So to me, the paper is great, but its kind of stopping too quickly, it feels like its only just tapping the surface. These kinds of ideas would be easy to test out and add to the papers.

If the authors add bit more meat to the paper, i.e. extra loss functions and hybrid-annealing (as discussed above), I would likely increase my score. Like I said, its just missing a little bit of filling to make it a great paper.

**Questions:**

See Weaknesses.

---

> ### Author Response · Authors · 2024-11-23
> **Author Responses(1/2)**
>
> Thanks for your time and effort in reviewing our paper! Please find our responses to your comments below. We will be happy to answer any further questions you may have.
>
> > **W1**: it would be nice to have a few more choices of losses, taking inspiration let say from f-divergences (Reverse-KL, JSD, etc.)
> > **W2**: there may be more than just zero-forcing and zero-avoiding to the key properties of loss functions hence why studying more losses would be helpful
>
> Following your suggestion, we have further developed five novel divergence-based loss functions, including forward and reverse $\chi^2$ distance, total variation, symmetric KL divergence, and Jensen-Shannon divergence. We conducted experiments on the bit-sequence generation task and observed that losses with the same zero-forcing/zero-avoiding properties lead to similar behaviors (see Tables 1, 2, and 3 below). This finding suggests that zero-forcing and zero-avoiding are the primary properties and the four representative losses discussed in our paper effectively capture their impacts. We have included the new loss functions and experimental results in **Appendix F** in the revision.
>
> It is very interesting to thoroughly explore other properties and effects of regression losses within the realm of $f$-divergence and beyond. We consider this an important topic for future research. We believe that the systematic framework presented in this work—comprising a unified framework that generalizes the training objectives of GFlowNets by identifying five key components, as well as the two-way connection between $f$-divergence and the regression loss function $g$—will be highly beneficial.
>
> P.S. The commonly used quadratic loss corresponds to reverse KL divergence, while our proposed Linex(1) loss corresponds to forward KL divergence.
>
> Table 1: Five novel loss functions
> | Loss | Divergence | Zero-forcing | Zero-avoiding |
> | -- | -- |:--:|:--:|
> | $g(t)=\frac{1}{4}(e^{2t}-2t-1)$ | Forward $\chi^2$ |  | $\checkmark$ |
> | $g(t)=e^{-t}+t-1$ | Reverse $\chi^2$ | $\checkmark$ |  |
> | $g(t)=\frac{1}{2}\|t\|$ | Total Variance |  |  |
> | $g(t)=\frac{1}{2}(e^t+\frac{1}{2}t^2-t-1)$ | Symmetric KL | $\checkmark$ | $\checkmark$ |
> | $g(t)=\frac{1}{2}\int_{1}^t\log\frac{e^x+1}{2}dx$ | Jensen-Shannon |  |  |
>
> Table 2: The number of runs that find all modes within 250k steps, and the median of the steps before they find all modes.
> |  | Zero-forcing | Zero-avoiding | TB | DB | STB |
> |--|:--:|:--:|:--:|:--:|:--:|
> | Reverse KL (baseline) | $\checkmark$ |  | $1/5$, $\ \ \ -\ \ \ \ $ | $\underline{5/5}$, $13.4k$ | $4/5$, $\ 50.6k\ $ |
> | Reverse $\chi^2$ | $\checkmark$ |  | $0/5$, $\ \ \ -\ \ \ \ $ | $0/5$, $\ \ -\ \ \ $ | $0/5$, $\ \ \ -\ \ \ $ |
> | Forward KL |  | $\checkmark$  | $\underline{5/5}$, $\ 98.0k\ $          | $\underline{5/5}$, $10.8k$         | $\underline{5/5}$, $\ 20.3k\ $          |
> | Forward $\chi^2$ |  | $\checkmark$  | $\underline{5/5}$, $\ \mathbf{80.3k}$ | $\underline{5/5}$, $\ \mathbf{8.1k}\ $ | $\underline{5/5}$, $\ \mathbf{10.2k}$ |
> | Hellinger |  |  | $\underline{5/5}$, $111.2k$ | $\underline{5/5}$, $11.7k$ | $\underline{5/5}$, $\ 55.9k\ $ |
> | Total Variation |  |  | $1/5$,  $\ \ \ -\ \ \ \ $ | $\underline{5/5}$, $47.1k$ | $2/5$, $\ \ \ -\ \ \ \ $ |
> | Jensen-Shannon |  |  | $4/5$, $162.2k$ | $\underline{5/5}$, $12.8k$ | $3/5$, $165.2k$ |
> | Shifted-Cosh | $\checkmark$ | $\checkmark$  | $4/5$, $\ 92.2k\ $ | $0/5$, $\ \ -\ \ \ $ | $\underline{5/5}$, $\ 90.0k\ $ |
> | Symmetric KL | $\checkmark$ | $\checkmark$  | $4/5$, $122.2k$ | $\underline{5/5}$, $13.7k$ | $\underline{5/5}$, $\ 27.5k\ $ |
>
> Table 3: The Spearman correlation between $P_T$ and $P_R$ over a test set (the higher the better). The failed runs where modal collapse happened are eliminated.
> |  | Zero-forcing | Zero-avoiding | TB | DB | STB |
> |--|:--:|:--:|:--:|:--:|:--:|
> | Reverse KL (baseline)$| $\checkmark$ |  | $\underline{0.8081}(\pm0.0159)$| $0.7907(\pm0.0175)$ | $\underline{0.8088}(\pm0.0169)$|
> | Reverse $\chi^2$ | $\checkmark$ |  | $\underline{0.8074}(\pm0.0129)$| - | $\underline{0.7899}(\pm0.0166)$|
> | Forward KL |  | $\checkmark$ | $0.7421(\pm0.0216)$ | $0.7464(\pm0.0107)$ | $0.7517(\pm0.0246)$ |
> | Forward $\chi^2$ |  | $\checkmark$ | $0.7507(\pm0.0174)$ | $0.7266(\pm0.0178)$ | $0.7439(\pm0.0126)$ |
> | Hellinger |  | | $0.7454(\pm0.0021)$ | $0.7580(\pm0.0132)$ | $0.7711(\pm0.0190)$ |
> | Total Variation |  | | $\underline{0.7893}(\pm0.0144)$| $0.7266(\pm0.0178)$ | - |
> | Jensen-Shannon |  | | $\underline{0.7852}(\pm0.0256)$| $0.7542(\pm0.0046)$ | $0.7640(\pm0.0213)$ |
> | Shifted-Cosh | $\checkmark$ | $\checkmark$ | $\mathbf{0.8122}(\pm0.0145)$ | $\mathbf{0.8213}(\pm0.0094)$| $\mathbf{0.8132}(\pm0.0149)$ |
> | Symmetric KL | $\checkmark$ | $\checkmark$ | $\underline{0.7908}(\pm0.0235)$| $0.7630(\pm0.0097)$ | $\underline{0.7886}(\pm0.0227)$|

---

> ### Author Response · Authors · 2024-11-23
> **Author Responses(2/2)**
>
> > **W3**: it would be nice to let say consider hybrid methods with some kind of annealing. For example, why not use Linex(1) for its fast convergence to a large number of nodes, before then transitioning to shifted-cos for higher rewards around those now-discovered modes.
>
> Thank you for your kind suggestion. It is quite an interesting idea! However, after our initial attempt, we decided not to include it in this paper and instead consider it as potential future work. There are two main reasons for this decision.
>
> First, the annealing loss did not perform as well as we had expected during our experiments. We suspect this is because the convergence point of different losses, i.e., the best approximation of the target distribution with respect to distinct divergence measures, can vary significantly, especially in complex real-world tasks. A simple example is that the best Gaussian approximation of a mixture of Gaussians w.r.t reverse KL is to fit the dominant peak, while that w.r.t forward KL tends to cover the whole support. Therefore, a more in-depth exploration is required to determine how to handle this process effectively.
>
> Second, achieving faster convergence is not the primary motivation for using zero-avoiding losses. In many real-world scenarios, the focus may be on enabling the trained GFlowNet to provide a diverse range of candidates. Considering this, it may be more preferred to offer different options for different desires than presenting a single seemingly "optimal" solution.
>
> Nonetheless, it remains intriguing to investigate whether it is possible to achieve the "best of both worlds" or even the "best of all worlds" with hybrid annealing strategies, and we plan to leave that for future research.
>
>
> We hope our responses fully address your concerns. If so, we wonder if you could kindly consider raising your rating score? We will also be happy to answer any further questions you may have. Thank you very much!

---

> > ### Comment · Reviewer_8npT · 2024-11-23
> > **Revision score**
> >
> > Thank you, this addresses a large portion of my requests. The paper is more detailed and better off this way. I am updating my score from 6/10 to 8/10.

---

> > > ### Author Response · Authors · 2024-11-25
> > >
> > > Thank you very much for your response, which help siginifcantly improve our paper! We really appreciate your time and effort in reviewing our paper.

---

### Official Review · Reviewer_EhxP · 2024-11-04

**Soundness:** 3
**Presentation:** 3
**Contribution:** 3
**Rating:** 8
**Confidence:** 3

**Summary:**

This paper presents a novel theoretical finding for GFlowNets regarding their objective function. Using f-divergence theories, they connect existing objectives of GFlowNets and show that they are special cases of the squared loss. They design a new loss structure that combines both properties together: (1) zero forcing (as considered in existing losses) and (2) zero avoiding, which compensates for exploration. Their new loss function seems to have empirical benefits.

**Strengths:**

1. The paper is well-written and easy to follow.


2. The theories are insightful.

**Weaknesses:**

1. The empirical results seem to be weak; they are only varied in synthetic tasks

**Questions:**

Areas for improvement and suggestions:

1. Making a literature connection with f-GAN, which also uses f-divergence in GANs, might be insightful to readers.


2. Include a discussion connecting with off-policy exploration methods. Are your loss and off-policy search orthogonal? Which means, is your loss function combined with an off-policy method (e.g., local search) better than the TB loss combined with an off-policy method?


3. It's good to see the categorization of prior GFlowNet works. Can you include this recent work [1] that uses genetic search as an off-policy method for training GFlowNets and provide some discussion?



[1] Hyeonah Kim et al., "Genetic-guided GFlowNets for Sample Efficient Molecular Optimization," NeurIPS 2024.

---

> ### Author Response · Authors · 2024-11-23
> **Author Responses(1/1)**
>
> Thanks for your time and effort in reviewing our paper! Please find our responses to your comments below. We will be happy to answer any further questions you may have.
>
> > **W1**: The empirical results seem to be weak; they are only varied in synthetic tasks.
>
> We emphasize that the tasks we conducted are popular and are widely adopted in the GFlowNets literature, e.g., [1][2][3][4]. In addition, the tasks are actually very challenging with high state and action dimensions.
>
> For instance, the molecule generation task is a challenging real-world application. The reward is a prediction of the binding energy of a molecule to a particular protein target sEH. There are up to $10^{16}$ valid states and between $100$ to $2000$ actions depending on the state.
>
> The bit-sequence generation task is also difficult. Under the configuration we choose ($n=120$, $k=8$), there are more than $10^{36}$ valid states and a range of $256$ to $3840$ actions, as the sequence is generated in a non-autoregressive manner.
>
> > **Q1**: Making a literature connection with f-GAN, which also uses f-divergence in GANs, might be insightful to readers.
>
> The original training objectives of both GFlowNets and GANs are theoretically linked to the reverse KL divergence. The f-GAN framework allows for the use of a broader class of divergence measures, specifically $f$-divergences, in training generative samplers. Similar approaches have also been applied to other algorithms, including VAE, VI, DPG, and DPO.
>
> Inspired by these efforts, we established the connection between $f$-divergence and the regression loss function $g$ within the training objectives of GFlowNets, based on which we derive novel loss functions for GFlowNets from various divergence measures. To demonstrate the effectiveness of these new loss functions, we conducted experiments across three different tasks: hyper-grid generation, bit-sequence generation, and molecule generation.
>
> Following your suggestion, we have revised the relevant paragraphs in **Section 2** in the revision.
>
> > **Q2**: Include a discussion connecting with off-policy exploration methods. Are your loss and off-policy search orthogonal? Which means, is your loss function combined with an off-policy method (e.g., local search) better than the TB loss combined with an off-policy method?
>
> Our loss functions and off-policy exploration methods are almost orthogonal.
>
> Firstly, our proposed loss functions, when combined with different exploration strategies, provide valid training objective functions, in the sense that the target distribution is perfectly matched if and only if the loss becomes zero.
>
> Secondly, our experiments in three different environments adopting forward policy and $\epsilon$-noisy forward policy as the exploration strategy, have shown the robustness of our analysis on different $g$ functions concerning the deviation of $\mu$ from the desired one in Theorem 4.1.
>
> Consequently, our loss functions can, in principle, be integrated with existing GFN training methods, including off-policy exploration strategies like local search, and be expected to preserve the exploration/exploitation features in such cases.
>
> To better explain this in the paper, we have included the above discussion at the end of **Section 4** in the revision.
>
> > **Q3**: It's good to see the categorization of prior GFlowNet works. Can you include this recent work [1] that uses genetic search as an off-policy method for training GFlowNets and provide some discussion?
>
> Thank you for providing the latest related work in the area. We have included it in our revision (**Section 4.1** and **Appendix A.2**).
>
> We hope our responses fully address your concerns. If so, we wonder if you could kindly consider raising your rating score? We will also be happy to answer any further questions you may have. Thank you very much!
>
> [1]Tiapkin, et al. "Generative flow networks as entropy-regularized rl." ICAIS 2024.
>
> [2]Bengio, et al. "Flow network based generative models for non-iterative diverse candidate generation." NeurIPS 2021
>
> [3]Malkin, et al. "Trajectory balance: Improved credit assignment in gflownets." NeurIPS 2022
>
> [4]Madan, et al. "Learning gflownets from partial episodes for improved convergence and stability." ICML 2023

---

> ### Author Response · Authors · 2024-11-25
> **Reminder to Reviewer EhxP**
>
> Dear Reviewer,
>
> Thank you for your time and effort in reviewing our paper.
>
> We hope our response has adequately addressed your concerns. If you feel that our rebuttal has clarified the issues raised, we kindly ask you to consider adjusting your score accordingly. Should you have any further questions or need additional clarification, we would be more than happy to discuss them with you.
>
> Thank you once again for your valuable feedback.
>
> Best regards,
>
> Authors

---

> > ### Comment · Reviewer_EhxP · 2024-11-25
> >
> > The rebuttal clarified my concerns. Score is updated. Thanks.

---

> ### Comment · Reviewer_EhxP · 2024-11-26
> **Last comment before camera ready**
>
> This is a good paper, but there still exist some minor points that can make it more professional.
>
> Before submitting the camera-ready version (no need to revise this in discussion period), please revise the references; some of them have already been published in the venue but are still marked as arXiv preprints.
>
> **Tip**: Do not rely on citation systems like Google Scholar (they are often suboptimal and not up-to-date); try to manually create a BibTeX file on your own (with strict rules). I have listed below the papers that have been published yet are noted as arXiv in your paper:
>
>
> ---
>
> Gflownet foundations --> JMLR
>
> Order-preserving gflownets --> ICLR
>
> Extreme q-learning: Maxent rl without entropy --> ICLR
>
> Generative flow networks assisted biological sequence editing --> NeurIPS
>
> Amortizing intractable inference in large language models --> ICLR
>
> Learning energy decompositions for partial inference of gflownets --> ICLR
>
> Pessimistic backward policy for gflownets --> NeurIPS
>
> Learning to scale logits for temperature-conditional gflownets --> ICML
>
> Local search gflownets --> ICLR
>
> Qgfn: Controllable greediness with action values --> NeurIPS
>
> Gflownets and variational inference --> ICLR
>
> Generative augmented flow networks --> ICLR
>
> Amortizing intractable inference in diffusion models for vision, language, and control --> NeurIPS
>
> Diffusion generative flow samplers: Improving learning signals through partial trajectory optimization --> ICLR
>
> Distributional gflownets with quantile flows --> TMLR

---

> > ### Author Response · Authors · 2024-11-26
> >
> > Thank you very much for raising your rating and your kind advice! We will be sure to update the references as suggested.

---

> > ### Author Response · Authors · 2024-11-29
> >
> > We have updated the references in our latest revision. Thank you for your kind advice!

---

### Official Review · Reviewer_Xe2i · 2024-11-07

**Soundness:** 2
**Presentation:** 3
**Contribution:** 2
**Rating:** 6
**Confidence:** 4

**Summary:**

This paper presents a novel framework for GFlowNet objective functions, unifying existing training algorithms and clarifying key components. By establishing a connection between objective functions and divergence measures, it offers valuable insights into designing effective training objectives. The authors investigate key regression properties—zero-forcing and zero-avoiding—and propose three new loss functions (Linex(1), Linex(1/2), and Shifted-Cosh) to balance exploration and exploitation. Extensive experiments on benchmarks, including hyper-grid, bit-sequence, and molecule generation, show that these losses outperform the common squared loss in convergence speed, diversity, quality, and robustness.

**Strengths:**

This paper introduces a systematic framework for designing regression losses in GFlowNet training, linking each loss to specific divergence measures for targeted properties. Resulting in three new losses—Shifted-Cosh, Linex(1/2), and Linex(1)—that enhance exploration and exploitation balance.

**Weaknesses:**

Broader exploration of other potential divergence-based losses would offer a more comprehensive understanding of the effects of different divergence properties on GFlowNet training. Although the novelty lies in extending GFlowNet loss functions, there are similar attempts in reinforcement learning and generative models. Although the paper derives theoretical properties of zero-forcing and zero-avoiding, it lacks direct theoretical comparison with existing GFlowNet training algorithms.

**Questions:**

Could the authors clarify if they’ve noticed stability shifts in higher-dimensional or complex tasks and if adjustments might bolster robustness? Additionally, what drives the choice of a limited set of losses—are there theoretical or practical reasons for omitting other f-divergences, like Hellinger? Lastly, insights into each loss function’s hyperparameter sensitivity and effects on convergence guarantees would further clarify their resilience and adaptability.

---

> ### Author Response · Authors · 2024-11-23
> **Author Responses(1/2)**
>
> Thanks for your time and effort in reviewing our paper! Please find our responses to your comments below. We will be happy to answer any further questions you may have.
>
> > **W1**: Broader exploration of other potential divergence-based losses would offer a more comprehensive understanding of the effects of different divergence properties on GFlowNet training.
>
> > **Q2**: Additionally, what drives the choice of a limited set of losses—are there theoretical or practical reasons for omitting other f-divergences, like Hellinger?
>
> Our primary contribution lies in establishing a systematic framework for analyzing and designing loss functions for GFlowNets, rather than focusing on specific loss functions. With a theoretical understanding of the zero-forcing and zero-avoiding properties, we propose novel representative loss functions to demonstrate their effects.
>
> Following your suggestion, we have further developed five novel divergence-based loss functions, including forward and reverse $\chi^2$ distance, total variation, symmetric KL divergence, and Jensen-Shannon divergence. We conducted experiments on the bit-sequence generation task and observed that losses with the same zero-forcing/zero-avoiding properties lead to similar behaviors (see Tables 1, 2, and 3 below). This finding suggests that zero-forcing and zero-avoiding are the primary properties and the four representative losses discussed in our paper effectively capture their impacts. We have included the new loss functions and experimental results in **Appendix F** in the revision. Further exploring other important divergence properties of loss functions would be an interesting future research topic.
>
> Regarding the Hellinger distance: It is a special case of $\alpha$-divergence when $\alpha=0.5$, based on which we derived Linex(1/2). We have adjusted the caption of **Table 2** in our revision for better clarification.
>
> Table 1: Five novel loss functions
> | Loss | Divergence | Zero-forcing | Zero-avoiding |
> | -- | -- |:--:|:--:|
> | $g(t)=\frac{1}{4}(e^{2t}-2t-1)$ | Forward $\chi^2$ |  | $\checkmark$ |
> | $g(t)=e^{-t}+t-1$ | Reverse $\chi^2$ | $\checkmark$ |  |
> | $g(t)=\frac{1}{2}\|t\|$ | Total Variance |  |  |
> | $g(t)=\frac{1}{2}(e^t+\frac{1}{2}t^2-t-1)$ | Symmetric KL | $\checkmark$ | $\checkmark$ |
> | $g(t)=\frac{1}{2}\int_{1}^t\log\frac{e^x+1}{2}dx$ | Jensen-Shannon |  |  |
>
> Table 2: The number of runs that find all modes within 250k steps, and the median of the steps before they find all modes.
> |  | Zero-forcing | Zero-avoiding | TB | DB | STB |
> |--|:--:|:--:|:--:|:--:|:--:|
> | Reverse KL (baseline) | $\checkmark$ |  | $1/5$, $\ \ \ -\ \ \ \ $ | $\underline{5/5}$, $13.4k$ | $4/5$, $\ 50.6k\ $ |
> | Reverse $\chi^2$ | $\checkmark$ |  | $0/5$, $\ \ \ -\ \ \ \ $ | $0/5$, $\ \ -\ \ \ $ | $0/5$, $\ \ \ -\ \ \ $ |
> | Forward KL |  | $\checkmark$  | $\underline{5/5}$, $\ 98.0k\ $          | $\underline{5/5}$, $10.8k$         | $\underline{5/5}$, $\ 20.3k\ $          |
> | Forward $\chi^2$ |  | $\checkmark$  | $\underline{5/5}$, $\ \mathbf{80.3k}$ | $\underline{5/5}$, $\ \mathbf{8.1k}\ $ | $\underline{5/5}$, $\ \mathbf{10.2k}$ |
> | Hellinger |  |  | $\underline{5/5}$, $111.2k$ | $\underline{5/5}$, $11.7k$ | $\underline{5/5}$, $\ 55.9k\ $ |
> | Total Variation |  |  | $1/5$,  $\ \ \ -\ \ \ \ $ | $\underline{5/5}$, $47.1k$ | $2/5$, $\ \ \ -\ \ \ \ $ |
> | Jensen-Shannon |  |  | $4/5$, $162.2k$ | $\underline{5/5}$, $12.8k$ | $3/5$, $165.2k$ |
> | Shifted-Cosh | $\checkmark$ | $\checkmark$  | $4/5$, $\ 92.2k\ $ | $0/5$, $\ \ -\ \ \ $ | $\underline{5/5}$, $\ 90.0k\ $ |
> | Symmetric KL | $\checkmark$ | $\checkmark$  | $4/5$, $122.2k$ | $\underline{5/5}$, $13.7k$ | $\underline{5/5}$, $\ 27.5k\ $ |
>
> Table 3: The Spearman correlation between $P_T$ and $P_R$ over a test set (the higher the better). The failed runs where modal collapse happened are eliminated.
> |  | Zero-forcing | Zero-avoiding | TB | DB | STB |
> |--|:--:|:--:|:--:|:--:|:--:|
> | Reverse KL (baseline)$| $\checkmark$ |  | $\underline{0.8081}(\pm0.0159)$| $0.7907(\pm0.0175)$ | $\underline{0.8088}(\pm0.0169)$|
> | Reverse $\chi^2$ | $\checkmark$ |  | $\underline{0.8074}(\pm0.0129)$| - | $\underline{0.7899}(\pm0.0166)$|
> | Forward KL |  | $\checkmark$ | $0.7421(\pm0.0216)$ | $0.7464(\pm0.0107)$ | $0.7517(\pm0.0246)$ |
> | Forward $\chi^2$ |  | $\checkmark$ | $0.7507(\pm0.0174)$ | $0.7266(\pm0.0178)$ | $0.7439(\pm0.0126)$ |
> | Hellinger |  | | $0.7454(\pm0.0021)$ | $0.7580(\pm0.0132)$ | $0.7711(\pm0.0190)$ |
> | Total Variation |  | | $\underline{0.7893}(\pm0.0144)$| $0.7266(\pm0.0178)$ | - |
> | Jensen-Shannon |  | | $\underline{0.7852}(\pm0.0256)$| $0.7542(\pm0.0046)$ | $0.7640(\pm0.0213)$ |
> | Shifted-Cosh | $\checkmark$ | $\checkmark$ | $\mathbf{0.8122}(\pm0.0145)$ | $\mathbf{0.8213}(\pm0.0094)$| $\mathbf{0.8132}(\pm0.0149)$ |
> | Symmetric KL | $\checkmark$ | $\checkmark$ | $\underline{0.7908}(\pm0.0235)$| $0.7630(\pm0.0097)$ | $\underline{0.7886}(\pm0.0227)$|

---

> > ### Comment · Reviewer_Xe2i · 2024-11-25
> >
> > Thank you, this is helpful and I increase my score.

---

> > > ### Author Response · Authors · 2024-11-25
> > >
> > > Thank you very much for your response, which help siginifcantly improve our paper! We really appreciate your time and effort in reviewing our paper.

---

> ### Author Response · Authors · 2024-11-23
> **Author Responses(2/2)**
>
> > **W2**: Although the novelty lies in extending GFlowNet loss functions, there are similar attempts in reinforcement learning and generative models.
>
> While our work was inspired by these attempts in other areas, we are advancing beyond those efforts in several ways.
>
> Firstly, we introduced a novel framework that unifies the different training objective functions of GFlowNets. This framework enables us to identify the key components of these objective functions.
>
> Secondly, we established a two-way connection between the $f$-divergence and the regression loss function $g$, which allows us not only to derive $g$ from a well-known $f$, but also to analyze any arbitrary $g$ using the corresponding $f$ (for example, the shifted-cosh loss and its related divergence).
>
> Thirdly, by utilizing the regression loss function $g$ instead of directly minimizing the $f$-divergence, our method can be effectively applied to off-policy or even offline training settings.
>
> To better connect with this line of work and to highlight our novel contributions in comparison to existing efforts, we have revised **Section 2** in our revision.
>
>  > **W3**: Although the paper derives theoretical properties of zero-forcing and zero-avoiding, it lacks direct theoretical comparison with existing GFlowNet training algorithms.
>
> Our proposed method is orthogonal to almost all existing GFlowNet training algorithms. Indeed, one of our main contributions is the development of a unified framework that identifies five key components involved in GFlowNet training algorithms: backward policy, training objectives, parameterization mapping, sampling/resampling weights, and regression loss. Most existing algorithms have primarily focused on all these components except for regression loss, making our work the first to investigate this critical yet often overlooked aspect. As a result, our proposed loss functions can, in principle, be integrated with most existing GFlowNet training methods, while still preserving their exploration and exploitation features.
>
> To provide a better clarification, we included a comparison of our theoretical results with previous findings in **Section 2**, along with a discussion on the compatibility of our methods with various exploration strategies at the end of **Section 4** in the revision.
>
> > **Q1**: Could the authors clarify if they’ve noticed stability shifts in higher-dimensional or complex tasks and if adjustments might bolster robustness?
>
> We appreciate the reviewer's observation on this important issue. In fact, tackling higher-dimensional or complex tasks is still an open problem and is yet fully addressed.
>
> Empirically, we increased the dimension in the hyper-grid environment in our experiments (from $20^4$ states to $20^5$ states). We observe that the baseline always fails to fit the distribution, while our proposed losses remain robust in most of the cases. These results have been included in **Section 5.1** in the revision.
>
> Studying the scalability and robustness of GFlowNet training algorithms is a very interesting topic for future research.
>
> > **Q3**: Lastly, insights into each loss function’s hyperparameter sensitivity and effects on convergence guarantees would further clarify their resilience and adaptability.
>
> From a theoretical aspect, although all the $f$-divergences and $g$ functions are convex, there are differences in the smoothness and strongly-convexity that may influence the optimal selection of hyperparameters. Empirically, we follow the training configurations of prior work, e.g., [1][2], in our experiments. Our results indicate that the proposed loss functions are not sensitive to the choice of hyperparameters. Our choice of hyperparameters and other experimental details can be found in **Appendix E**.
>
> We hope our responses fully address your concerns. If so, we wonder if you could kindly consider raising your rating score? We will also be happy to answer any further questions you may have. Thank you very much!
>
> [1]Tiapkin, et al. "Generative flow networks as entropy-regularized rl." ICAIS 2024.
>
> [2]https://github.com/GFNOrg/gflownet

---

### Public Comment · ~Eliezer_de_Souza_da_Silva1 · 2024-11-27
**Some relevant references**

We appreciated the authors' contributions to this domain and wanted to highlight some missing relevant references.

In our recent work, published at NeurIPS 2024, we investigated the properties of $\alpha$-divergences (including forward and reverse Kullback-Leibler (KL), R\'enyi-$\alpha$, and Tsallis-$\alpha$ divergences) in the context of training GFlowNets. In particular, we also presented the trade-offs between zero-forcing and zero-avoiding behaviors for this family of learning objectives.

While this paper approaches the problem differently, we believe that acknowledging our work would provide additional context for understanding the theoretical and practical aspects of GFlowNet training.

Additionally, we noticed that Heiko Zimmermann et al.'s work exploring the relationship between GFlowNets and Variational Inference (VI), appears to be missing from the related discussion. Including this reference would enrich the paper's perspective and be appropriate for a complete overview of recent works.

Thank you for considering our feedback. We hope this fosters a richer discussion and continued progress in the field!

References:
1. Tiago Silva, Eliezer de Souza da Silva, and Diego Mesquita. *On Divergence Measures for Training GFlowNets.* NeurIPS, 2024. [Link](https://openreview.net/forum?id=N5H4z0Pzvn)
2. Heiko Zimmermann et al. *A Variational Perspective on Generative Flow Networks.* Transactions on Machine Learning Research, 2023. [Link](https://openreview.net/forum?id=AZ4GobeSLq)

---

> ### Author Response · Authors · 2024-11-29
>
> Thank you for pointing to relevant references. We have now included them in our latest revision.
>
> To connect your studies with ours, the forward KL divergence corresponds to our proposed Linex(1) loss, while the Tsallis-$\alpha$ divergence with $\alpha=0.5$ corresponds to our proposed Linex(1/2) loss (up to a multiplicative constant). Following our analysis, the performance gain from using such divergence measures may stem from the zero-avoiding or non-zero-forcing properties. Additionally, by utilizing the $g$ function instead of directly optimizing the $f$-divergence, our method can accommodate balance conditions beyond just TB, and allows for off-policy exploration strategies.
>
> We agree that approaching this problem from different perspectives enhances our understanding of both the theoretical and practical aspects of GFlowNet training. We really appreciate your comments.

---

### Meta-Review · Area_Chair_9rBq · 2024-12-20

**Metareview:**

In the paper, the authors addressed the issue of choosing regression loss in Generative Flow Networks (GFlowNets). They demonstrated a connection between regression losses and specific divergence measures. This connection enables the systematic design and evaluation of regression losses, tailored to the unique properties of their corresponding divergence measures.

All the reviewers agree that the theoretical results are novel and insightful. The presentation of the paper is clear and easy to follow. After the rebuttal, most of the remaining concerns of the reviewers are addressed and all the reviewers are happy with the current stage of the paper.

While there are some concerns about the limited experimental results, I believe that the current novelty and contribution of the paper are sufficient for ICLR. Therefore, I recommend accepting the paper at the current stage. The authors are encouraged to incorporate the feedbacks and comments of the reviewers into the camera-ready version of their paper.

**Additional Comments On Reviewer Discussion:**

Please refer to the meta-review.

---

### Decision · Program_Chairs · 2025-01-22

Accept (Spotlight)